

# Development of operational decision support tools for mechanized ski guiding using avalanche terrain modelling, GPS tracking, and machine learning

John Sykes[1,2], Pascal Haegeli[3], Roger Atkins[4], Patrick Mair[5], Yves Bühler[6,7]

[1] Simon Fraser University Department of Geography, Burnaby, BC, Canada
[2] Chugach National Forest Avalanche Center, Girdwood, AK, USA
[3] Simon Fraser University School of Resource and Environmental Management, Burnaby, BC, Canada
[4] Canadian Mountain Holidays, Banff, AB, Canada
[5] Harvard University Department of Psychology, Cambridge, MA, USA
[6] WSL Institute for Snow and Avalanche Research SLF, Davos, Switzerland
[7] Climate Change, Extremes and Natural Hazards in Alpine Regions Research Centre CERC, Davos, Switzerland

*Correspondence to*: John Sykes (john_sykes@sfu.ca)

**Abstract.** Snow avalanches are the primary mountain hazard for mechanized skiing operations. Helicopter and snowcat ski guides are tasked with finding safe terrain to provide guests with enjoyable skiing in a fast-paced and highly dynamic and

complex decision environment. Based on years of experience, ski guides have established systematic decision-making practices that streamline the process and limit the potential negative influences of time pressure and emotional investment. While this expertise is shared within guiding teams through mentorship, the current lack of a quantitative description of the process prevents the development of decision aids that could strengthen the process. To address this knowledge gap, we collaborated with guides at Canadian Mountain Holidays (CMH) Galena Lodge to catalogue and analyze their decision-making

process for the daily run list, where they code runs as green (open for guiding), red (closed), or black (not considered) before heading into the field. To capture the real-world decision-making process, we first built the structure of the decision-making process with input from guides, and then used a wide range of available relevant data indicative of run characteristics, current conditions, and prior run list decisions to create the features of the models. We employed three different modelling approaches to capture the run list decision-making process: Bayesian Network, Random Forest, and Extreme Gradient Boosting. The

overall accuracies of the models are 84.6%, 91.9 %, and 93.3% respectively, compared to a testing dataset of roughly 20,000 observed run codes. The insights of our analysis provide a baseline for the development of effective decision support tools for backcountry avalanche risk management that can offer independent perspectives on operational terrain choices based on historic patterns or as a training tool for newer guides.



## 1 Introduction

Snow avalanches are a complex and dynamic natural hazard, responsible for an average of approximately 140 recorded fatalities annually in North America and Europe (Colorado Avalanche Information Center, 2024; Jamieson et al., 2010; Techel et al., 2016). The majority of these avalanche fatalities are backcountry recreationists, and the avalanche is commonly triggered by a member of the victim's party (Schweizer and Lütschg, 2001). Terrain selection is the primary tool for managing avalanche risk when travelling in the backcountry. A wide range of factors need to be considered to select appropriate terrain, including

current avalanche conditions, slope incline, forest density, aspect, elevation, and potential for overhead hazards or terrain traps. The dynamic nature of avalanche hazard conditions and sheer number of influences on avalanche terrain hazard make choosing appropriate terrain challenging.

Due to the complexity of the terrain selection process, there is a long-standing desire to provide recreationists with decision-making aids for making better informed decisions about when and where to travel in the backcountry. Early tools such as the

seminal Graphical Reduction Method (Munter, 1997), the Stop-Or-Go method (Larcher, 1999), the SnowCard (Engler and Mersch, 2001), the NivoTest (Bolognesi, 2000), or the Avaluator (Haegeli, 2010; Haegeli and McCammon, 2007) provided users with relatively simple, analog workflows to combine information on conditions (mainly represented by the danger rating published by an avalanche warning service) with terrain information (primarily slope incline) to assess the severity of different routes. Current trip planning tools such as WhiteRisk (https://whiterisk.ch/) or Skitourenguru (https://www.skitourenguru.ch/)

are modern incarnations of the original approaches that take advantage of recent developments in avalanche terrain modelling to describe the severity of avalanche terrain in more detail. While these tools can be effective for general recreationists, their simplicity, particularly their focus on the public avalanche danger rating limits their value for more complex decision-making contexts such as professional guiding or advanced amateur recreation. In the case of mechanized ski guiding in Canada, the decision-making process includes an added layer of operational considerations, which further increases complexity.

Based on decades of practical experience, the mechanized skiing industry has developed a structured and iterative process to select terrain that is appropriate for skiing on a daily basis (Israelson, 2015). The decision-making process consists of four major components. First, guides assess current avalanche hazard conditions and produce an avalanche forecast that is relevant for the entire guiding tenure. Second, they create a run list which determines which ski runs within their tenure are available for guiding based on the current conditions. Based on the run list and operational conditions for the day (e.g., weather

conditions, snow quality, skills and preferences of guests, flying logistics), the third step is selecting which ski runs will be used for the day, which is carried out by lead guides in collaboration with the guiding team. The selection of ski runs is an ongoing process throughout the day which can be altered by changing avalanche or weather conditions. Finally, many runs can be skied in a variety of ways with different terrain characteristics and exposure to avalanche hazard. It is the responsibility of the guide of each group to select an appropriate ski line based on evaluation of slope scale avalanche conditions, ski quality,

and operational considerations.





The practice of creating a daily run list helps guiding teams to get on the same page for the day and establishes a list of potential terrain that has been deemed appropriate for the day's conditions. Individual ski runs can be coded open for guiding (green), closed for guiding (red), or not-considered (black). Black codes essentially represent non-decisions (i.e., default) describing the situation when guides do not think the run is worth discussing during their roughly 15-minute run coding meeting. The

reasons for not discussing a run include insufficient snow coverage on a run, the run being too far away given the current flying conditions, the terrain being obviously too hazardous to consider for the current conditions, or too much uncertainty for making an informed decision. Hence, the causes of a run not being coded clearly differ from a run being coded red versus green. In addition, guides' personal references and biases can impact whether a run is coded as black. The process of coding runs during the morning meeting prior to going skiing gives the opportunity for a consensus-based decision process and helps limit

emotional and time pressures that can impact decision-making in the field. HeliCat Canada, the association of mechanized skiing operations in Canada, identifies daily run lists as a crucial component of avalanche risk management practices (HeliCat Canada, 2024).

Quantitatively describing the run list coding process in a way that provides insight and offers added value for participating operations requires sophisticated model approaches that can consider the wide range of relevant factors and capture the nuanced

nature of these decisions. Prior research has used regression analysis as a method for capturing decision-making processes (Sterchi et al., 2019; Thumlert and Haegeli, 2018), which assumes that the decision to open or close a run can be represented as a linear combination of factors. These approaches provided useful starting points for capturing the complexity of guiding decisions but are limited by the modelling methods. Purely data-driven machine learning methods, such as using self-organizing maps for grouping runs based on run code patterns (Sterchi and Haegeli, 2019), have also shown promise but are

prone to detecting spurious relationships, and the black box nature of the algorithms makes them difficult to understand and trust.

Recent advances in artificial intelligence and machine learning have led to the development of a wide range of different algorithms which show promise for both examining guide decisions in more sophisticated ways and developing meaningful operational decision support tools. Bayesian networks (BN) offer an attractive alternative to the existing methods due to their

ability to use expert knowledge to model complex decision processes (Fenton and Neil, 2019). Decision tree base methods, such as Random Forests (RF) and Extreme Gradient Boosting (XGB), are also attractive for modelling complex decision-making tasks due to their ability to automatically account for complex relationships within the data and their track record of producing accurate predictions in a variety of modelling domains (Breiman, 2001; Chen and Guestrin, 2016). Furthermore, improvement of methods for interpreting the output of machine learning models has led to a greater ability to understand what

is going on under the hood of black box models, which makes them more transparent and has the potential to improve trustworthiness in implementing these tools in operational settings (Molnar, 2022).





The objective of this paper is to describe the run list coding process at a mechanized skiing operation using BN, RF, and XGB approaches and discuss their potential for the design of operational decision support tools for the mechanized skiing industry. We explore the factors that influence run list decisions and the relationships within the decision-making process. The empirical 95 foundation of the decision-making models is based on seven seasons of operational data (winter 2015/16 - winter 2022/23) as well as high resolution avalanche terrain modelling. We test and compare the performance of the decision-making models as predictive tools and use interpretable machine learning methods to understand the inner workings of the black box models. The insights from this study lay the foundation to collaborate with guiding operations to create real world decision support tools that capture historic decision-making patterns with the potential for integration into guide training and daily operational 100 decision-making practices.

## 2 Methods

Capturing the critical factors for the run list coding process at an operation requires a variety of different data sets, which can be grouped into factors that characterize the terrain within each ski run, current conditions, and operational factors and constraints. This section first introduces the study area and data sets that we used to capture the run list decision-making 105 process. It then discusses the three modelling methods and our approach to model evaluation in detail. A table of all variables included in the decision support models is in Appendix A, including a description of the variable, a histogram of the variable distribution, and how it is applied in each model. The code necessary to reproduce our analysis are available at https://doi.org/10.17605/OSF.IO/6DHMX (Sykes et al., 2024), all code is written in the R program for statistical computing (R Core Team, 2024). Due to the large number of data sources and variables included in the present analysis, it is not possible 110 to describe every processing step in complete detail within the constraints of this paper. However, interested readers are encouraged to reach out to the corresponding author for more details.

### 2.1 Study area

Canadian Mountain Holidays (CMH) Galena Lodge is a mechanized skiing operation located in the Selkirk Mountains near Trout Lake, BC, Canada (Figure 1). The Selkirk Mountains have a transitional snow climate, prone to persistent weak layers 115 of surface hoar and faceted layers associated with icy crusts (Haegeli and McClung, 2007). Most of the terrain in the CMH Galena tenure is forested, but there are also high alpine zones with glaciated ski runs. Within their roughly 1,200 km$^2$ tenure are 295 predefined individual ski runs (Figure 1), which are individually coded as green, red, or black each morning. For this research we only included ski runs that are completely within the operating boundaries of CMH Galena's tenure, where we have collected at least 10 GPS tracks over the study period (see Section 2.2.2), and where information about operational

considerations was available (see Section 2.2.3). This results in an analysis dataset for 192 ski runs, which are highlighted in yellow in Figure 1.

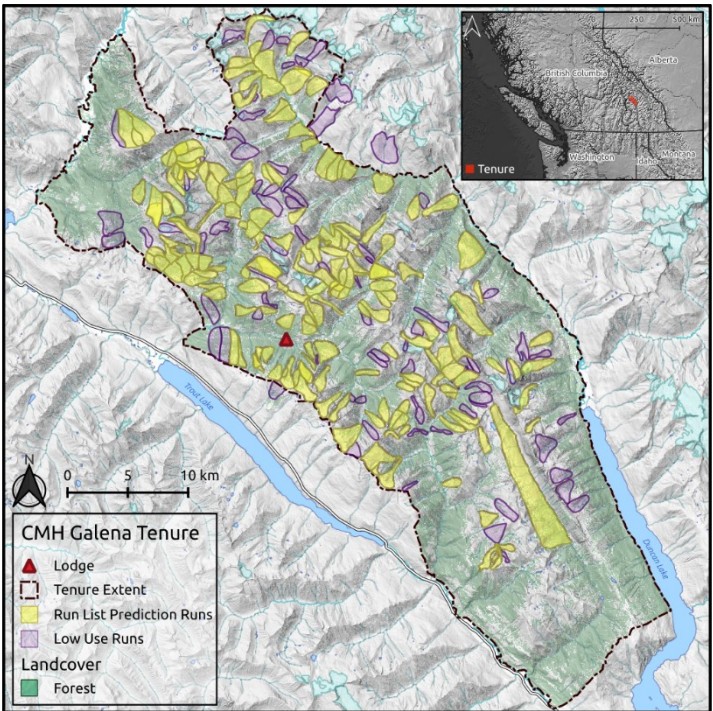

**Figure 1: Canadian Mountain Holidays Gelena tenure, showing the lodge location and ski runs included in the analysis. Each run is coded using the run list during the daily morning guides meeting.**

**2.2 Run characteristics**

To characterize the terrain in the CMH Galena tenure, we modelled avalanche start zones and runout zones using state-of-the-art avalanche terrain modelling methods. The output of these models describes the terrain across the entire tenure but to better understand the characteristics of the terrain where guides commonly travel, we focused on the characteristics of raster pixels within a 20 m buffer of GPS tracks that have been collected by the research team to record guides' terrain choices since the

2015/16 winter season. Based on discussions with guides, we learned that guides only consider the 'most conservative line' within the run during the run list coding process, therefore on runs that are heavily used we applied a clustering approach to identify the most conservative ski line from the collected GPS tracks to extract relevant raster pixels. To include the terrain characteristics in decision support tools we calculated summary statistics to represent the terrain on each ski run. In addition to the physical terrain characteristics, we also included operational factors for each ski run that play a role in the decision-

making process. The following paragraphs explain each of these steps in more detail.



### 2.2.1 Avalanche terrain modelling

The data we used to characterize avalanche terrain at CMH Galena include elevation, forest cover, exposure to potential avalanche release areas (PRA), and avalanche runout zones. Elevation data came from a SPOT 6 satellite stereophotogrammetry 5 m DEM and forest cover was estimated using land cover classification of Rapid Eye 5 m satellite
imagery (Sykes et al., 2022). We used a potential release area (PRA) model to estimate the extent and size of avalanche start zones based on slope angle, aspect, curvature, roughness, and forest density (Bühler et al., 2013, 2018, 2022; Sykes et al., 2022). To quantify exposure to overhead hazard, we used the large-scale hazard indication modelling approach described by Bühler at al. (2022) including the avalanche dynamics model RAMMS (Christen et al., 2010) to simulate the runout distance, velocity, impact pressure, and flow height for avalanches originating from all 111,937 identified PRA polygons.

We simulated PRAs and overhead hazard for two different avalanche scenarios (Figure 2): A frequent scenario targeting smaller storm snow slab avalanches that are commonly encountered (Sykes et al., 2022), and a large scenario that is intended to capture deeper and more connected avalanches that are more typical of periods with active persistent weak layers. For the frequent scenario we use the 10-year return period parameters to identify potential release area polygons and to increase the size for the large scenario we used the 30-year parameters from prior research conducted by Bühler et al., 2022. The size of
potential avalanche release areas is typically larger for the large avalanche scenario, but the extent to which the release area polygons differ between the two scenarios depends on the local terrain characteristics. For the frequent scenario we use a release depth of 0.5 m for the RAMMS simulations and for the large scenario we use a release depth of 1 m. The release depth values are based on discussion with local guides and targeted around the type of avalanche activity we aim to capture with the simulations.

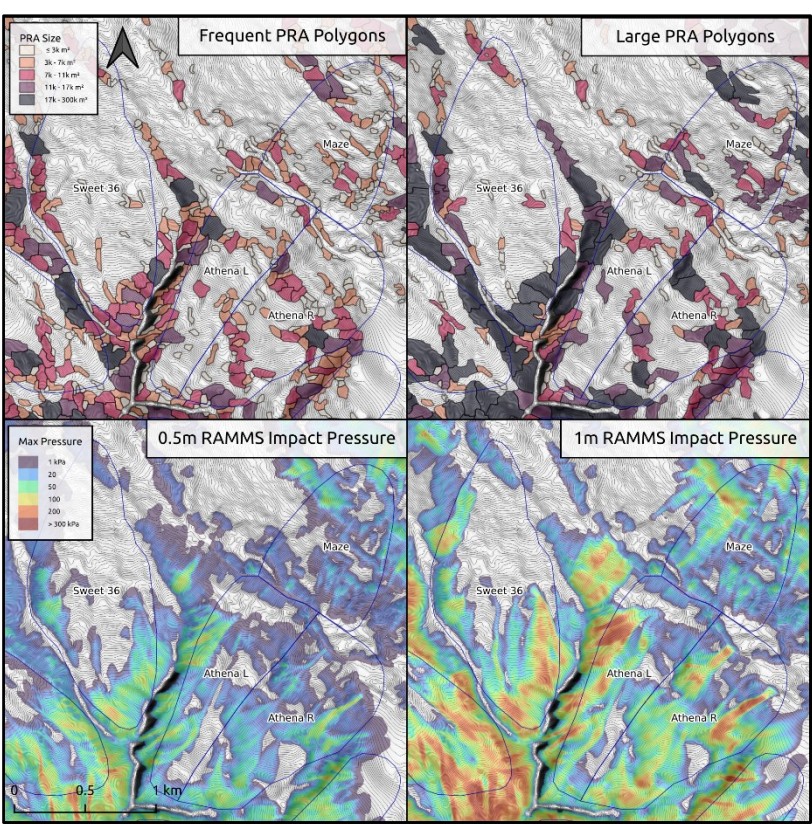


**Figure 2: Comparison of PRA polygons and runout impact pressure for frequent and large runout simulations.**

### 2.2.2 GPS tracking

Starting in the winter of 2015/2016 the Simon Fraser University Avalanche Research Program has collaborated with several
mechanized ski guiding operations in Western Canada to collect high-resolution information on the terrain skied. The location
information was collected with custom-designed GPS trackers which recorded participating guides' positions every 4 sec over
the course of a week (Thumlert and Haegeli, 2018). At CMH Galena, the research team has collected 15,111 GPS tracks over
seven winter seasons (2020/21 season is missing due to COVID-19 restrictions).

We leverage the GPS tracking data in our run list decision-making model by using the GPS track coordinates to extract terrain
characteristics for each run. This method is more accurate than using the predefined run polygons (Figure 3) because it focuses
the spatial extent of the terrain characterization to only the portion of the run polygon that is actually skied. Since the most
conservative ski line matters the most for opening or closing a run, we used a clustering approach to further refine the portion
of the run that we use to characterize the terrain on heavily used runs, which we defined as having 50 or more GPS tracks over
the data collection period (n = 65).





To identify the most conservative line within the available GPS tracks associated with a ski run polygon, we grouped the tracks
using fuzzy analysis clustering, a probabilistic variant of the k-medoid clustering approach described in Chapter 4 of Kaufman
and Rousseeuw (2005) and implemented in the fanny function of the cluster package in R (Maechler et al., 2022). In
comparison to hard or deterministic clustering, fuzzy clustering calculates membership probabilities for each datapoint to
describe how likely they belong to a particular cluster. This allows the method to provide better insight into datasets where the
differences between clusters are more gradual (Kaufman and Rousseeuw, 2005). The distance matrix used for the clustering
was a combination of two normalized distance matrices: one for the geographic location represented by the start and end point
of the GPS tracks (i.e., coordinates of landing and pickup locations) and one for the terrain characteristics of the tracks, which
included the 95[th] percentiles of slope incline, PRA area for the frequent and large scenario, runout pressure for the frequent
and large scenarios, as well as the proportion of the track in forested terrain. Terrain characteristics that likely did not exhibit
a multimodal distribution (as tested with Hartigans' dip test from the diptest R package by Maechler, 2021) were eliminated
from the terrain characteristics distance matrix. Based on our initial explorations, the default values for the weight of the terrain
characteristics in the overall distance matrix and the fuzzy parameter that determines the crispness of the cluster solutions were
0.15 and 1.7 respectively. For each ski run, we calculated solutions for 2 to 10 clusters and selected be best solution based on
the average silhouette width, one of the commonly used measures for assessing how well the data points are represented by
their clusters. Subsequently, the most conservative line within the selected cluster solution was identified by examining the
distributions of the terrain characteristics of DEM raster cells touched by the GPS tracks associated with the different clusters.
To minimize the influence of outliers, only GPS tracks with a cluster membership probability higher than 0.75 were included
in these assessments. The selected cluster solutions and most conservative lines were verified by CMH Galena guides, and if
necessary, the algorithm was rerun with slightly modified parameter values to produce a more realistic solution. Figure 3
presents the identified ski lines for several runs to illustrate the output of the clustering algorithm.


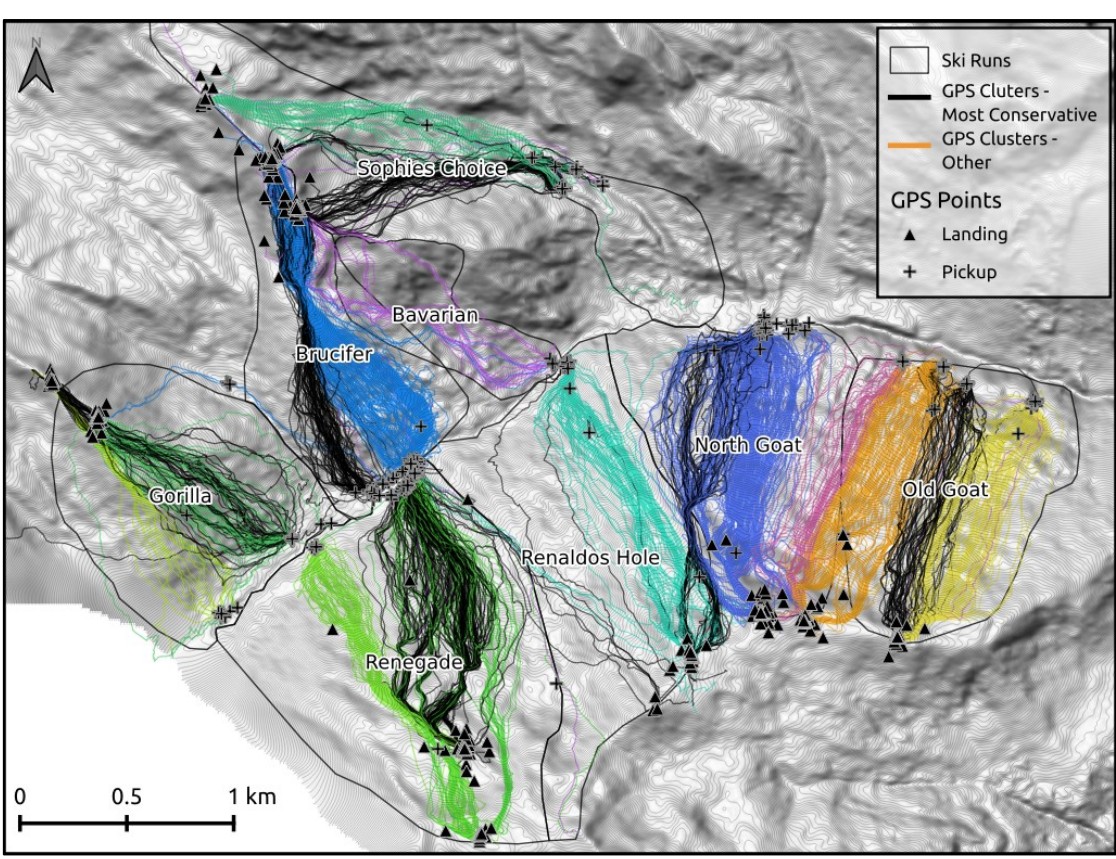

**Figure 3: Example of GPS clustering approach, with most conservative cluster of GPS tracks shown with black lines and other GPS track clusters shown by color coded lines.**

### 2.2.3 Summarizing terrain characteristics

Since the unit of decision-making for the run list coding is an individual ski run, we needed to simplify the terrain information for each run into a single number summary for each terrain characteristic. We carried this out by extracting the mean, median, 95th percentile, and maximum values for PRA area, runout depth, runout impact pressure, runout velocity, and slope incline based on the values of all raster cells within a 20 m buffer of the relevant GPS tracks. For the PRA and runout terrain layers we calculated these summaries for the output of both the frequent and large avalanche scenarios. In addition, we calculated the percentage of each run that was within PRA, covered by forest, and the proportion of each run within each cardinal aspect. To further characterize the aspect of the runs we calculated the average northness of each run, which uses a cosine transformation to determine the degree of northern exposure of a run ranging from -1 (south) to 1 (north) (Olaya, 2009). Since separate avalanche hazard assessments are produced for alpine, treeline and below treeline, we also calculated the percentage of each





run in the different elevation bands using elevation thresholds from local guides (alpine > 2250 m, treeline 2250 – 1850 m, below treeline < 1850 m). Finally, we extracted the maximum and minimum elevation of each run.

### 2.2.4 Operational considerations

The final component of run characterization aims to capture the operational perspective of each run. One key source of this information was the work of Wakefield (2019) who developed a ski run characterization survey to capture the guides' personal perception and operational knowledge of their skiing terrain (Wakefield, 2019). The collected information contains a wealth of knowledge from CMH guides, but the present study only incorporates a limited subset including a) whether weak layers are intentionally managed by destroying them on the surface using skier traffic, b) whether a run fills a specific operational role (e.g., lunch run or destination run), c) the approximate flight distance required to access a ski run, d) the overall ski quality of the run, and e) the overall accessibility of the run and the landing zone for each run.

### 2.3 Current conditions

There is a multitude of condition factors that can impact run list coding at CMH Galena, but in this manuscript we present a relatively sparse model that focuses on the major decision drivers. We selected the variables based on the operational experience of Roger Atkins and looked for relationships within the data. In the absence of high-quality weather station data in our study area, we relied on field observations, lodge weather observations, and daily avalanche hazard assessments to capture the current conditions. In addition, we include the following daily-changing operational factors relevant to the daily run list coding: a) the percent of the tenure that was observed the prior day, b) how long it had been since the run had been skied last. In addition, c) whether the guiding program was on an exchange day, where guests change, guides teams are swapped out, and operational logistics such as transporting food and equipment to the lodge impact the daily operations. These factors help to incorporate real world operational considerations that have an impact on the decision-making process, which are independent from the terrain hazard or avalanche hazard conditions. The following sections describe how the condition variables were derived in detail.

### 2.3.1 Weather conditions

Snowfall loading rates are the most critical factor to determine the size and likelihood of avalanches. Therefore, we included three variables related to snow loading in our decision-making models: the height of new snow over the past 72 hours ($hn72$), 24 hours ($hn24$), and 12 hours ($h2d$) as recorded in the daily field observations and morning lodge observations from the guiding team. We also include the daily average height of snow ($hs$) observed in the field as a proxy for the overall snow coverage in the tenure. Additional weather factors captured from guides field observations include wind speed, sky cover, and





current precipitation rate. As an indicator of seasonal changes to operational practices and general mountain conditions we also included the time of season as an ordinal variable (early-winter, mid-winter, early-spring, and spring).

### 2.3.2 Avalanche conditions

To capture the guiding team's understanding of the avalanche hazard conditions, we extracted daily avalanche hazard ratings for each elevation band, avalanche likelihood and avalanche size information of the identified avalanche problems, and the strategic mindset from their morning assessments (Atkins, 2014; Statham et al., 2010, 2018). Since recent avalanche observations play a large role in avalanche hazard assessment, we also included the total number and maximum size of the avalanches observed within the CMH Galena tenure from the past 72 hours and the past week. We elected to separate avalanche likelihood and size information for persistent and non-persistent avalanche problems to capture the effect of different types of

avalanche problems on the decision-making process more precisely.

### 2.3.3 Run coding

The daily run list codes are the output variable for our decision-making models. While working with Roger Atkins to understand the CMH Galena run list coding process, we realized that transition periods are the most interesting and useful target for a decision-making model as they indicate a change in operational conditions from the status quo. However, these

transition periods are relatively infrequent, only accounting for roughly 11% of the run list codes in our data set, while runs remain red roughly 18%, remain green roughly 59%, and remain black in roughly 12% of run list codes. To maximize the utility of the decision support tools we constructed our target variable to explicitly highlight transitions from the prior day's run code. The run list target variable in our models includes five categories: 'closing', 'status black', 'status red', 'status green', 'opening'. We consider runs transitioning from 'green' to either 'red' or 'black' as the run 'closing'. Conversely, anytime a

run transitions from 'red' or 'black' to 'green' we consider it 'opening'.

We also included the run list coding from the prior day as a variable in our decision-making models. This captures the iterative nature of the run coding, where codes are updated daily based on prior observations and new information. Including the prior run list code as an anchor for the daily run list coding is realistic to the real-world decision-making process and allows us to explicitly identify periods of transition within the run coding.

### 2.4 Model development and evaluation

We tested three different statistical models to develop decision-support tools for the run list coding process: a Bayesian network (BN), random forest (RF), and extreme gradient boosting (XGB). The BN approach has the advantage of being explicitly grounded in expert understanding of the decision-making process. This type of model is new to the avalanche field but has been applied in other applications where uncertainty and dynamic conditions are integral to the decision-making context





(Fenton and Neil, 2019). We selected two ensemble-based machine learning approaches to test their potential for developing a decision support tool: Random Forest (RF) because it is a commonly used model across a variety of domains, including other applications in the avalanche field (Mayer et al., 2022; Richter et al., 2023), and Extreme Gradient Boosting (XGB) because it is well known as a state-of-the-art model for high predictive performance (Chen and Guestrin, 2016).

### 2.4.1 Bayesian network

Bayesian networks (BN), also known as belief networks or probabilistic graphical models, are a type of statistical model that are used to represent and analyze uncertain complex relationships among multiple variables that include both inputs and outputs (Scutari and Denis, 2021). The foundation of a BN is a directed acyclic graph (DAG), which illustrates variables as nodes and relationships between them as arcs. The graphical structure of a BN cannot contain any cycles, and nodes that are not connected by an arc are assumed to be conditionally independent given their parents (Fenton and Neil, 2019; Scutari and Denis,

2021). One major advantage of BNs over other types of modern machine learning algorithms is that the structure of the network can be constructed based on input from domain experts, which allows the network to take on a form which is authentic to real world decision-making thought processes. The quantitative foundation of BN are conditional probability tables (CPT), which can be estimated manually or based on observed data for each node. BNs have been applied in a variety of fields, including medical diagnosis and operational risk analysis (Fenton and Neil, 2019). Once a BN has been estimated, it can be used for a

variety of tasks related to probabilistic inference, prediction, and decision support, which make BNs well suited to our task. In this analysis we used the R packages bnlearn (Scutari, 2010) and gRain (Højsgaard, 2012) to fit and apply the BN.

The main driver for deciding what nodes to include in the BN and how to set arcs between nodes was the expert opinion of Roger Atkins, a long-time guide at CMH Galena. The primary objective of this step was to capture realistic patterns of decision-making in the arc pathways within the DAG of the BN. We then used the data described in the previous section to calculate

the conditional probability distributions of the BN based on the structure provided by the domain expert.

We constructed the DAG based on the thought process of using three different types of relationships to set arcs (Figure 4). First are arcs between run characteristic nodes, which represent the natural physical relationships in avalanche terrain and operational relationships in the guide survey nodes. Second are arcs between condition variables, which represent the relationships between observations and guides' assessments, which are roughly modelled after the theoretical foundation of

the conceptual model of avalanche hazard (Statham et al., 2018). Third are decision arcs that connect nodes that could have a direct impact on how a run is coded. Each of these three types of arcs are included in the BN for different purposes, but all are relevant for meaningfully model the decision-making process with a BN.

To reduce the complexity of building the BN and make it easier to understand, we elected to use categorical variables for all the nodes in the network. This required converting the numeric variables into categorical, which we undertook manually and

aimed to minimize the number of categories with very small proportions of the data to reduce the overall size of the conditional



probability table for the run list output node. Reducing the number of categories in each variable significantly reduces the computer processing time to apply the BN in a predictive capacity and lead to more accurate predictions. See Appendix A for a list of all variables and to compare the original numeric distribution to the categorized version of the variables.

**2.4.2 Machine learning approaches**

Both machine learning approaches are based on decision trees but differ in their specific implementation. A decision tree is a common statistical approach to classification where a simple tree structure is built to split data into leaves or nodes based on a training dataset that includes both feature values and the desired classification. The internal nodes of a decision tree represent a test on an individual feature in the data, with the branches descending from each internal node representing the outcome of the test (Breiman, 2001). The terminal nodes, or leaves of the tree, represent the classification prediction. One of the main

advantages of decision trees is that they automatically detect relationships within the data without the analyst needing to a priory specify them and naturally handle interactions among features (Kuhn and Johnson, 2013). However, when applied to predictive problems individual decision trees tend to overfit the sample observations, which means they do not tend to generalize well to observations outside of the training data set.

Random forests (RF) are an ensemble-based machine learning approach which uses hundreds of independent decision trees to

produce more accurate and generalizable predictions. Independent decision trees are fit by using a random subsample of the training data, using a process called bagging, and the feature used at each node within the tree is selected from a random subset of all the features available. These practices allow the individual trees within the RF to be substantially different from one another, which improves overall performance of the ensemble (Breiman, 2001). A majority voting scheme is used to determine the prediction of the RF, which means whichever classification level gets the most votes of all the trees becomes the overall

prediction.

While RF uses bagging and random sampling to create a forest of independent trees, Extreme Gradient Boosting (XGB) uses a technique called gradient boosting to sequentially build decision trees that correct the errors made by previous trees (Chen and Guestrin, 2016). Gradient boosting creates an ensemble of simple weak learners, defined as a simple classifier with performance slightly better than random chance, to form a strong learner, defined as a classifier that achieves arbitrarily good

accuracy, by optimizing a loss function when each new tree is fit. Essentially each subsequent weak learner in the ensemble focuses on the misclassified cases from prior weak learners to focus training on cases the model previously got wrong. This method allows XGB to produce classification models with reduced bias and variance which leads to better predictive performance. Compared to RF, XGB tends to build more complex models that capture more nuanced patterns in the data. Fitting XGB models typically requires more computer processing time and tuning of several model parameters to achieve

optimal results.



To fit the machine learning models to our dataset we included all variables from the BN and added additional variables related to the run list decision-making process as determined by conversations with expert guides and evaluated whether they were improving predictive performance. While the RF model easily adopted the categorical variables that we developed for the BN approach, they needed to be manually encoded using one-hot encoding before including them in the XGB method, using the

R package fastDummies (Kaplan, 2020). To ease interpretation of the XGB model we elected to use the native numeric representations of the run characteristic and conditions variables where possible. In addition, we tested treating ordered factors as both dummy coded variables and as ordered integers in the XGB model.

To tune the RF model parameters, we performed a grid search on the 'mtry' parameter, which determines how many variables are randomly sampled at each split in the decision trees using the R packages caret (Kuhn, 2008) and RandomForest (Liaw,

Andy and Wiener, Matthew, 2002). For the XGB model we used the 'gbtree' booster and carried out a more extensive process of tuning the 'nrounds', 'eta', 'gamma', 'max_depth', 'subsample', 'colsample_bytree', and 'min_child_weight' model parameters using the R packages caret (Kuhn, 2008) and XGBoost (Chen and Guestrin, 2016). We aimed to optimize overall accuracy and used the default 'softmax' objective function from the summary function 'multiClassSummary'. Our process for tuning the XGB model parameters required five steps; 1) roughly tune 'nrounds', 'eta', and 'max_depth' while limiting the

max value of 'nrounds' to 1000 to limit processing time of the tuning procedure and using default values for other parameters, 2) tune 'max_depth' and 'min_child_weight' using 'nrounds' values from 50 to 1000 using tuned 'eta' value from round 1 and defaults for other parameters, 3) tune 'colsample_bytree' and 'subsample' using tuned parameters for 'eta', 'max_depth', and 'min_child_weight' while using default parameter for 'gamma', 4) tune 'gamma' using 'nrounds' values from 50 to 1000 and tuned parameters for all other values, and finally 5) tune 'eta' and 'nrounds' a second time using tuned parameters for all other

inputs and testing a larger range of 'nrounds' values from 100 to 5000. This process is intended to sequentially tune parameters in batches to limit computer processing time while still testing a wide range of potential parameter combinations.

We tested both the RF and XGB models with and without class weights, which are intended to improve accuracy for imbalanced classification tasks. We used a class weights scheme based on the inverse proportion of the sample size, so that errors in classes with lower sample sizes are penalized more heavily than classes with larger sample sizes.

**2.4.3 Model evaluation**

To assess how well our decision-making models match real world decisions, we used each model as a predictive tool to classify whether runs would be coded as 'closing', 'status black', 'status red', 'status green', or 'opening' based on run characteristics and current conditions. We used 70% of our run list dataset to train the models and 30% to test the prediction accuracy. To compare the models, we used a multiclass confusion matrix. Specifically, we examined the overall accuracy and Cohen's

kappa, which are metrics for the overall performance of the classifier, and the sensitivity of individual classes to evaluate performance in greater detail. The overall accuracy is the proportion of cases where the model predicts the same run list code



as the CMH guides. Cohen's kappa measures how well the classifier performs compared to a model that simply predicted the most frequent class, also known as the no information rate. In addition to the confusion matrix approach, we calculated the area under the receiver operating curve (AUROC) for each model using the R package pROC (Robin et al., 2011). AUROC

considers the sensitivity and specificity of the model and evaluates the overall performance of the classifier, with an AUROC of 0.5 indicating random chance and 1.0 indicating a perfect classifier. Since our output variable has multiple classes, the AUROC is calculated using a one versus one approach, with each class considered as the positive case and compared against every possible pairwise combination of classes. The model AUROC is then calculated as the average of the pairwise AUROC values (Hand and Till, 2001; Robin et al., 2011).

To better understand the patterns captured in the RF and XGB models, we also looked at the feature importance for these models. Feature importance provides an assessment of which variables contribute to the classification task most strongly, which is determined by the mean decrease in the Gini coefficient (Breiman, 2001). This measures how much each feature contributes to homogeneity of the nodes in the decision trees, which leads to more accurate classification. To dig deeper into understanding the relative contribution of different features, we created Shapley additive explanations (SHAP) plots, which

are a more advanced method for interpretating black box machine learning models (Lundberg and Lee, 2017; Molnar, 2022). In addition to measuring which features strongly contribute to the classification task, SHAP also show how the range of values for each feature contributes to the classification (Flora et al., 2024). SHAP can be calculated for both the overall model and the individual levels of the classification. These methods help to visualize how the machine learning models create their classifications and provide some insight into the patterns that drive these black box models.

**3 Results**

**3.1 Bayesian network**

Our decision-making network aimed at capturing the daily run list coding at CMH Galena contains 24 nodes and 44 arcs (Figure 4). To fit the BN, we used 63,581 cases to train the network and kept 27,254 cases to test the accuracy of the BN. Overall, the network structure represents the complexity of the decision-making process by containing many potential

pathways to the run list node. This realistically represents the real-world decision-making process, where the driving factor for the coding of runs depends on a multitude of factors related to current conditions and run-specific characteristics.

**3.1.1 Input nodes – terrain characteristics and operational factors**

We included seven nodes in the BN that represent terrain characteristics from the avalanche terrain model output (light blue nodes). Potential avalanche release area size (*pra*) represents the 95$^{th}$ percentile start zone polygon size for the frequent scenario

within each ski run, which is categorized into four classes (0-10,000 m$^2$, 10,000-15,000 m$^2$, 15,000-20,000 m$^2$, > 20,000 m$^2$).





PRA size is aimed at capturing the high end of the distribution of avalanche release areas that could be triggered on the run. The percentage of the ski run that is within potential avalanche release areas (*pra_perc*) intends to capture the overall amount of exposure to areas where avalanche could be triggered along the run (0-25 %, 25-40 %, 40-55 %, 55-100 %). Runout size (*runout_size*) represents the 95th percentile impact pressure from the large avalanche simulation, which was included to

represent the high-end potential avalanche runout size, or overhead hazard, during periods where large avalanches are possible (0-50 kPa, 50-100 kPa, 100-150 kPa, > 150 kPa). Runout depth (*runout_depth*) is determined by taking the 95th percentile runout height for the frequent avalanche scenario, which captures the potential of terrain traps to cause deep burial in case of a relatively small human triggered avalanche (0-1 m, 1-1.5 m, 1.5-2 m, > 2 m). The node for run steepness (*slope*) represents the steepest portion of each run by using the 95th percentile of its slope angle distribution, which is then categorized into four

classes (0-35°, 35-40°, 40-45°, > 45°). We chose to use the 95th percentile value to capture the upper end of the distribution for PRA size, slope angle, runout size, and runout depth instead of the maximum value to minimize the potential for local DEM artifacts to give unrealistically high maximum values. To represent the elevation (*elev*) of a run we used all the elevation bands a run includes, so runs that cover multiple elevation bands include multiple elevation bands (alpine-treeline, alpine-treeline-below treeline, treeline-below treeline, below treeline). Forest cover (*forest*) was summarized based on the percentage

of raster cells within 20 m of GPS tracks that are forested and split into categories of 0-25 %, 25-50 %, 50-75 %, or 75-100 %. There are several inherent correlations among the run characteristics that need to be accounted for in the model with arcs. PRA size is connected by arcs to runout size and runout depth because the surface area of the start zone has a strong impact on potential avalanche size and burial depth. PRA percent is connected by an arc to forest cover percent (*forest*) because avalanche start zones tend to inhibit the growth of forests. Elevation band (*elev*) and runout size also have an arc connected to *forest*

because large avalanche paths and higher elevations both inhibit the growth of forests.

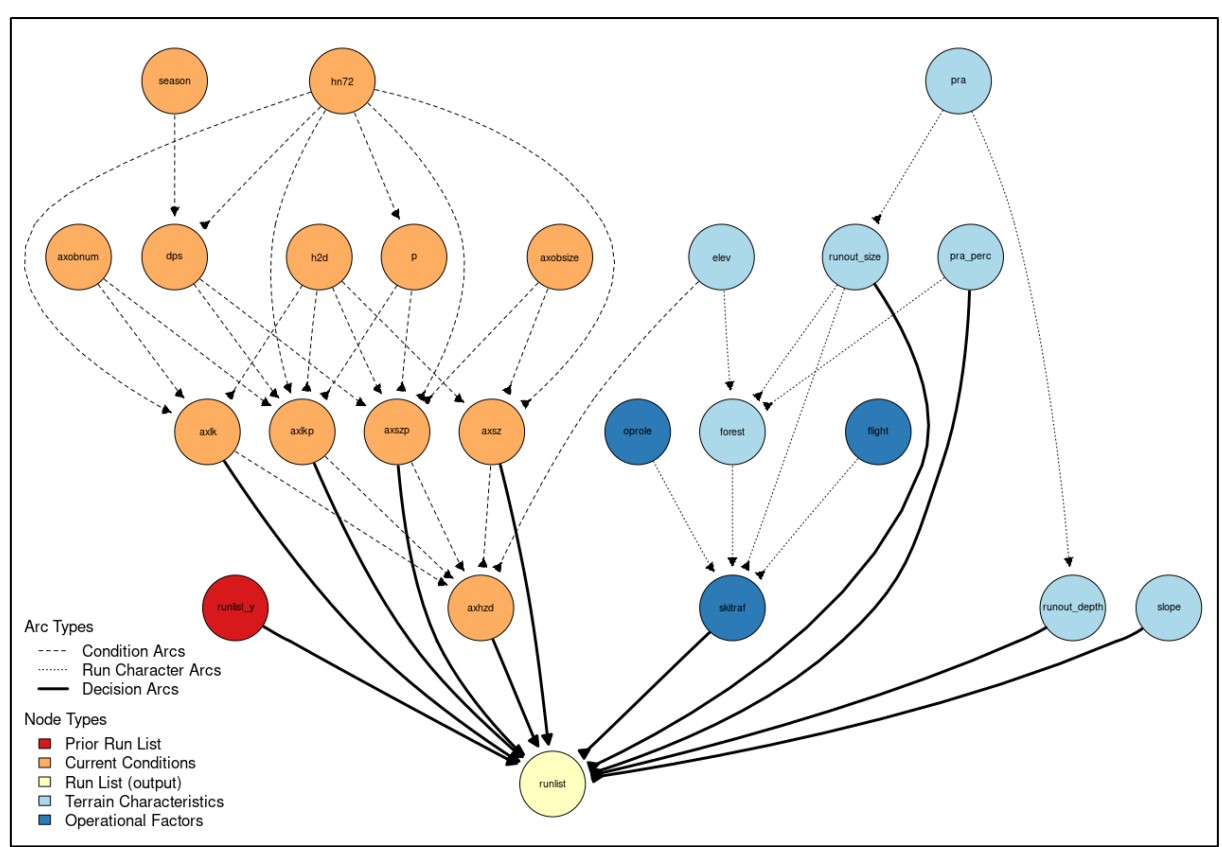

**Figure 4: Directed acyclic graph (DAG) for run list BN at CMH Galena. Arcs are defined based on the expert opinion of our collaborating guide as well as natural physical relationships of avalanche terrain characteristics.**

The operational factors included in the BN (dark blue nodes) are whether skier traffic is used to mitigate weak layer

development (*skitraf*), the flight distance from the lodge to the run (*flight*), and whether the run serves a specific operational

role (*oprole*). The nodes *oprole* and *flight* both have arcs connecting to skier traffic because runs that are maintained with skier

traffic tend to be in closer proximity to the lodge and serve a specific operational role because they can typically be used during

periods of elevated avalanche hazard. The skier traffic node also has input arcs from *forest* and *runout_size* because forest

cover can break up potential avalanche start zones into multiple smaller start zones, which are more suitable for this type of

mitigation, and runs with exposure to large overhead avalanche paths are typically not suitable for skier mitigation.

### 3.1.2 Input nodes – current conditions

Twelve nodes are included in the BN model to represent current avalanche conditions (orange nodes). These nodes include

both direct observations and guide assessments of the conditions. The relationships among these condition variables are driven

by physical principles and the avalanche hazard assessment process described by the conceptual model of avalanche hazard





(Statham et al., 2018). The primary weather condition variables in the BN are the height of new snow within 72 hours (*hn72*) and the height of new snow within 12 hours (*h2d*). These nodes are included in the model to represent the amount of new snow loading within the current storm and overnight respectively and naturally have arcs connected to non-persistent avalanche size (*axsz*), non-persistent avalanche likelihood (*axlk*), size of persistent avalanches (*axszp*), and likelihood of persistent avalanches (*axlkp*). In addition, *hn72* has arcs connected to the status of persistent (*p*) and deep persistent (*dps*) avalanche problems, which have values of 0 when the avalanche problem is not active and 1 when active. Time of season (*season*) is a secondary condition variable that is oriented towards the development of snowpack characteristics over the course of a winter season. *Season* is connected to the status of deep persistent avalanche problems, which tends to be less likely in the early winter and more likely in the mid-winter, early-spring, and spring. The number of avalanche observations (*axobnum*) and maximum size of avalanche observations (*axobsize*) within 72 hours from the guides' field observations represent their direct evidence of current avalanche activity. There are arcs connecting observed avalanche size to expected avalanche size for persistent and non-persistent avalanche problems, and from number of observed avalanches to expected avalanche likelihood for both persistent and non-persistent avalanche problems. The status of persistent and deep persistent avalanche problems are each connected with arcs to persistent avalanche likelihood and size. As described in the conceptual model of avalanche hazard (Statham et al., 2018), avalanche size and likelihood nodes for both persistent and non-persistent avalanche problems have arcs to daily maximum avalanche danger rating (*axhzd*) as they are the key determining factors of avalanche hazard. Since the daily maximum avalanche danger rating is specific to the elevation bands included in each run, an arc connects elevation band to avalanche hazard.

### 3.1.3 Output node

The target output node is run list, which captures the change in the run list status from the prior day using the classes *closing*, *status black*, *status red*, *status green*, and *opening*. This node has input arcs from avalanche size, avalanche likelihood, persistent avalanche size, persistent avalanche likelihood, avalanche hazard, runout size, runout depth, percent of PRA, slope angle, skier traffic mitigation, and the run list status from the prior day (*runlist_y*). By combining condition variables, run characteristics, and prior status we aimed to capture the interactions between the range of potential factors that drive the run coding decisions for different types of runs.

### 3.1.4 BN performance

The BN has an overall accuracy of 84.6 percent compared to the test cases with an area under the receiver operating curve (AUROC) of 0.87 and a kappa statistic of 0.74 (Table 1). The no information rate for the BN sample is 59.0 %, which is the class frequency of 'status green'. For the transition classes 'closing' and 'opening' the BN has a sensitivity of 27.8 % and 24.4 % respectively. For complete results of the confusion matrix for the BN model see Appendix B. The BN fitting process


does not include a method for class weighting. However, as an alternative to prioritize performance of the transition classes
we tested manually setting the classification threshold for 'closing' and 'opening' to 25% instead of simply selecting the class
with the highest probability. We found that the performance in transition classes improved substantially from 27.8% to 40.5%
for 'closing' and 24.4% to 34.7% for 'opening'. However, manually setting the classification threshold to improve sensitivity
for transition cases results in a decrease in overall accuracy and Cohen's kappa, from 84.6% to 81.7% and 0.74 to 0.70

respectively.

**Table 1: Accuracy metrics for three decision support tools using 30% of run list data for model evaluation.**

|  | BN | RF | XGB |
|---|---|---|---|
| **n features** | 23 | 42 | 58 |
| **Size of test dataset** | 27,254 | 20,899 | 20,898 |
| **AUROC** | 0.87 | 0.97 | 0.98 |
| **Overall accuracy** | 84.6 | 91.9 | 93.3 |
| **Kappa** | 0.74 | 0.87 | 0.89 |
| **'closing' sensitivity** | 27.8 | 70.5 | 72.0 |
| **'opening' sensitivity** | 24.4 | 50.2 | 56.8 |

## 3.2 Random Forest

### 3.2.1 Features included

To fit the RF model, we started with the same features that we included in the BN but added additional features to provide

more detailed information about the run characteristics and current conditions. Due to missing data in the additional features
for the RF, the overall dataset was slightly smaller than the BN, with 48,755 cases in the training data set and 20,899 in the
testing data set. The final set of features was tested by trial and error and evaluated against the accuracy metrics from the
testing data set. Our grid search for tuning the 'mtry' parameter resulted in a value of 9, which means that nine features were
randomly selected and tested at each split while growing the decision trees. We used the default value of 500 for the number

of trees in the RF. To account for the imbalance in the run list classification target variable, we used inverse proportional
weighting to penalize errors in the minority classes more heavily while training the model. This improved performance for the
transition periods *closing* and *opening*, which makes the model more useful as an operational decision support tool.

The additional terrain characteristics included in the RF model are 95[th] percentile PRA size for the large avalanche scenario
(*pra30y*), average PRA size along the run for the frequent (*pra_mean*) and large scenarios (*pra30y_mean*), 95[th] percentile

runout pressure for the frequent scenario (*runout_press*), 95[th] percentile runout height for the large scenario
(*runout_height_1m*), and the aspect (*aspect*) of the run with the highest proportion of raster cells within 20 m of GPS tracks.





Each of these variables was manually converted to a categorical variable for inclusion in the RF. These additional features help to capture the unique characteristics of each run based on the exposure to avalanche start zones and runout zones.

We also added several additional operational factors to the RF model to help capture the some of the more nuanced operational
considerations that can impact run list coding. Those features are the number of days since the run was last skied (*last_skied*), the overall quality of skiing on the run (*ski_quality*), overall accessibility of the run (*access_gen*), accessibility of the landing zone (*access_land*), and whether there was an exchange of guests or guides (*exchange*) taking place that could impact operational logistics. The features capturing number of days since the run was skied and guest exchange were manually converted to categorical variables, while the others originate from categorical variables from the guide perspective survey data.

To capture the current conditions in more detail, we also added additional features to the RF model focused on weather conditions, field observations, and avalanche hazard assessment. The weather condition features we added are the height of new snow in the past 24 hours (*hn24*), the wind speed observed from the field (*wind*), sky cover observed from the lodge in the morning (*sky*), and the current precipitation rate from the lodge in the morning (*precip*). Additional field observations include the total snowpack height (*hs*) observed in the field the day prior, the percent of the tenure observed on the prior day
(*perc_observed*), and the maximum size of avalanche observations over the past week (*axobs_sizeweek*). We also replaced the daily max avalanche hazard feature from the BN with avalanche hazard ratings for each elevation band (alpine – *alp_hzd*, treeline – *tl_hzd*, below treeline – *btl_hzd*). To capture the shared mindset of the guiding team, we included the strategic mindset (*mindset*) as a feature for the RF model. Finally, we removed the status of deep persistent avalanche problem and persistent avalanche problem from the RF model because the persistent likelihood and size features capture this information
implicitly, and the overall accuracy of the RF decreased with these avalanche problem features included.

### 3.2.2 Feature importance

The feature with by far the highest feature importance for the RF model is the run list from the prior day, which naturally emerges from the fact that in roughly 90 % of cases the run list code stays the same as the prior day (Figure 5). Features 2 through 7 by feature importance are all related to current conditions, with the most important features being strategic mindset,
new snow in past 24 hours, treeline hazard rating, overall height of snow, alpine hazard rating, and likelihood of persistent avalanches. The below treeline hazard rating is also ranked relatively highly in 15[th]. The remaining features that capture snow loading, three-day snow loading (*hn72*) and 12-hour snow loading (*h2d*), are ranked 19[th] and 21[st] by feature importance. The avalanche observation features that are most important are the percent of the tenure observed the prior day (*perc_observed*) and total number of avalanches observed over a 3-day period (*axobs_num72*) which are ranked 10[th] and 18[th].
Operational features with the highest feature importance are overall accessibility of the run (*access_gen*) ranked 8[th] and flight distance (*flight_dist*) in 9[th]. Other highly ranked operational features are the quality of the skiing experience on the run (*ski_quality*) in 17[th] and whether a run is maintained by skier traffic (*skier_mitig*) in 26[th]. The least important features in the





RF model are operational role (*op_role*) in 41st, which designates runs as having a specific value to operational logistics beyond physical characteristics, and whether the guiding program is on an exchange day (*exchange*) in 42nd.

The terrain characteristics that are highest ranked by feature importance occupy positions 11 through 14 and are the aspect of the run (*aspect*), runout height for the frequent scenario (*runout_height*), runout height for the large scenario (*runout_height_1m*), and runout pressure for the frequent scenario (*runout_press*). Features related to PRA are generally ranked lower compared to those runout features, with the most important PRA features being PRA size for the frequent scenario (*pra*) ranked 20th, mean size of PRA for the large scenario (*pra30y_mean*) ranked 22nd, mean size of PRA for the frequent

scenario (*pra_mean*) ranked 25th, and percent of raster cells in PRA areas (*pra%*) ranked 27th.

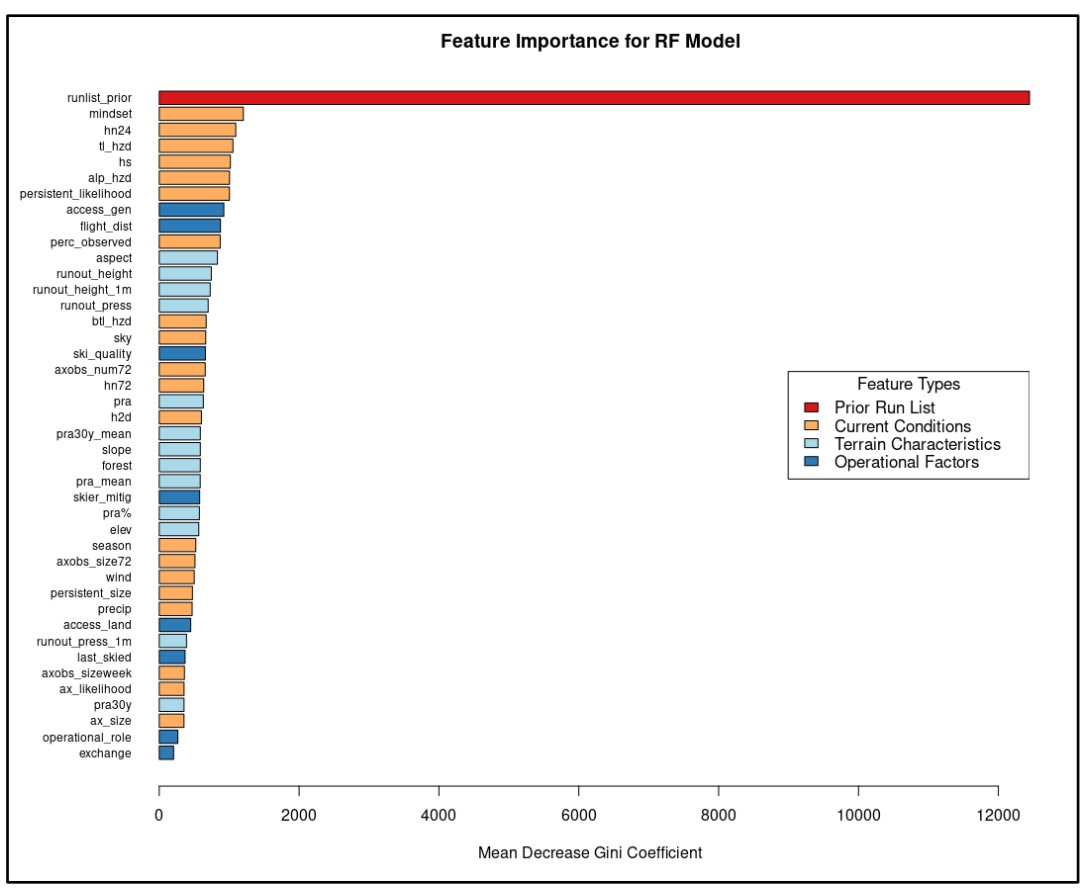

**Figure 5: Feature importance for RF model for all classes of the run list output variable color coded by the feature type.**

### 3.2.3 RF performance

The RF overall accuracy is 91.9% with an AUROC of 0.97 and a kappa statistic of 0.87. The no information rate for the RF

data set is 57.4 %, which is the class frequency of status green. The improvement in predictive performance of the RF model





over the BN is 7.3 percentage points in overall accuracy, 0.10 for AUROC, and 0.13 for kappa. The sensitivity for the 'closing' and 'opening' classes for the RF is 70.5 % and 50.2 % respectively, which is an improvement of 42.7 and 25.8 percentage points respectively compared to the BN with default classification thresholds. For complete results of the confusion matrix for the RF model see Appendix B. Tuning the RF model without class weighting resulted in a higher overall accuracy by 0.8

percentage points and a higher Cohen's kappa by 0.01. However, the class weighted RF has higher sensitivity for closing and opening classes by 15.1 and 9.7 percentage points respectively.

### 3.3 Extreme Gradient Boosting

### 3.3.1 Features included

For the XGB model we used the same features as the RF. However, since the XGB model requires numeric features, we

reverted to the original numeric data structure wherever possible. For features that did not originate as numeric variables, we converted ordered factors into integers and unordered factors into dummy coded categorical features. The only features that were dummy coded were the categorical strategic mindset and the run list from the day prior. For elevation, we switched from categorizing which elevation bands are part of each run in the RF to calculating the percentage of each run in the alpine (*perc_alp*), treeline (*perc_tl*), and below treeline (*perc_btl*) elevation bands. We also included the maximum (*elev_max*) and

minimum elevation (*elev_min*) for each run. To accurately capture the influence of slope aspect we switched from using a categorical variable representing the majority aspect in the RF to calculating the average northness of each run (*northness*). This is relevant for the run list decision-making context because southerly slopes and northerly slopes can have dramatically different snowpack structures due to the influence of solar radiation.

Sample sizes for the XGB training and testing data sets are almost identical to the RF, with 48,756 cases for training and

20,898 cases for testing. The grid search procedure to optimize the XGB model parameters resulted in 'nrounds' of 4,400, 'eta' of 0.05, 'max_depth' of 6, 'gamma' of 0.05, 'colsample_bytree' of 0.4, 'min_child_weight' of 2, and 'subsample' of 1.

### 3.3.2 Feature importance

Figures 6 and 7 visualize the feature contributions by plotting the SHAP values for all possible outcomes of the run list target variable (Figure 6) as well as for the individual transition classes *closing* and *opening* (Figure 7). The features on the SHAP

plots are ordered on the y-axis by their feature importance and the x-axis shows the SHAP value. The top three features by feature importance for the overall classification are the dummy coded features that represent the status of the run from the prior day. This indicates that the prior days run list code is the strongest predictor of the current days run list code. Since the run list code only changes in roughly 10% of cases it makes sense that these features have the strongest contribution to predictive performance. The points along the x-axis for each feature show the distribution of feature values ranging from low (yellow) to



high (purple). The distribution of the points is shown by the shape of the bee swarm plot, with a higher density of feature values shown as a thicker section of the point distribution. For the top two features in Figure 6, prior run list green and prior run list red, we see that high feature values have high SHAP values, which indicates that the prior run list code being green or red (coded as 1 for dummy variables) has a very strong contribution to the run list classification. In contrast the prior run list being black, shown by a high feature value, has a much lower SHAP value. This means that the prior run list being black contributes much less to the run list classification than green or red. Since the run list code black is considered a non-decision that can have a variety of reasons unrelated to current conditions or terrain characteristics, it makes sense that this run list code would contribute less to the XGB model predictions.

Of the remaining top 20 features shown in Figure 7 there is a mix of operational factors, terrain characteristics, and current conditions. The operational factors included in the top 20 features are number of days since the run was last skied, flight distance to the run, and percent of tenure observed the prior day. For both number of days since last skied and percent of terrain observed higher values have a stronger contribution to the overall classification. Whereas lower values of flight distance have a higher contribution to the classification.

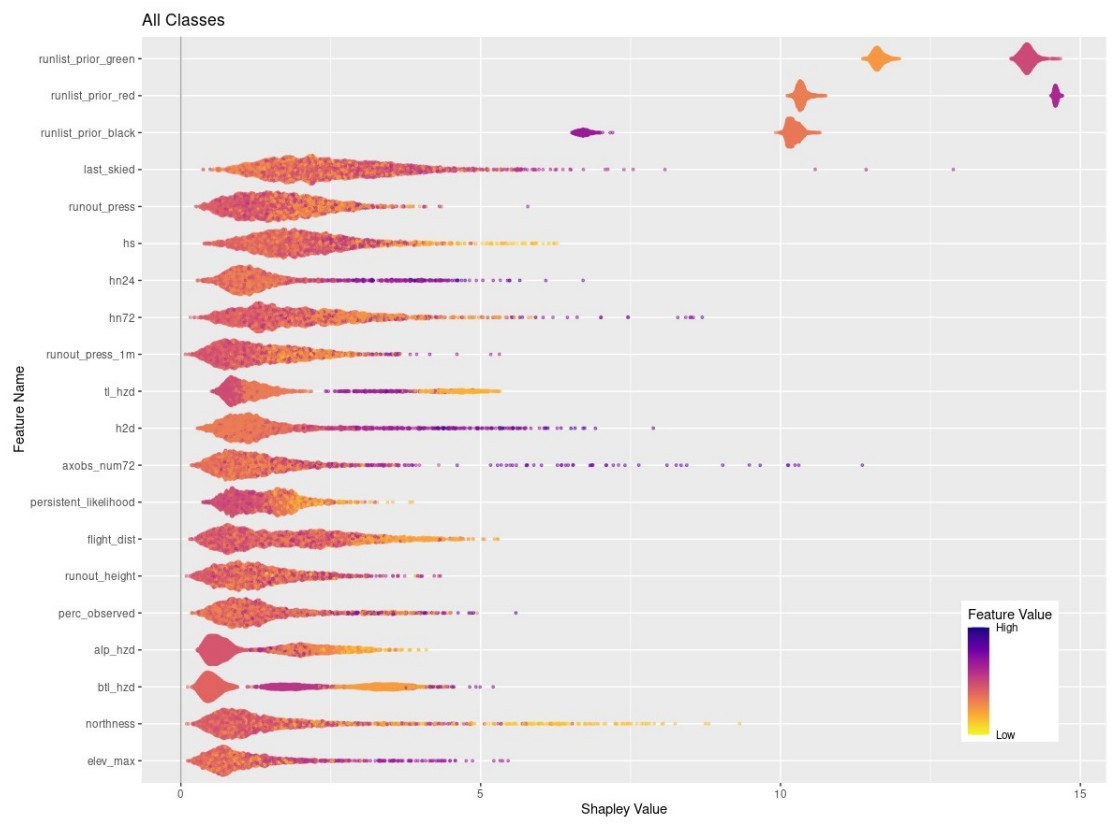




**Figure 6: SHAP plot for XGB model with the top 20 features on the y-axis ordered by feature importance and x-axis showing the**
**SHAP value. The color-coded points show the distribution of the individual features with hotter colors indicating high feature values**
**and cooler colors indicating low feature values.**

The terrain characteristics with the highest contribution to the XGB classification are runout impact pressure from the frequent
avalanche scenario, runout impact pressure from the large avalanche scenario, runout height from the frequent avalanche
scenario, degree of northness of the run, and the maximum elevation of the run. In general, lower values of runout pressure for
the frequent and large scenario contribute more strongly to the classification, whereas higher values of runout height have a
stronger contribution. Runs with low values of northness (i.e. run with southern aspects) have higher contributions, along with
runs that start at higher elevations.

Nine out of the 20 top features for the overall classification represent the current conditions. The total snowpack height as well
as all three snow loading features (72-hour, 24-hour, and 12-hour) are included. The avalanche hazard rating for all three
elevation bands is also included, with a general trend that low or high values have stronger contribution compared to
intermediate values. This in intuitive because high avalanche hazard or low avalanche hazard both represent hazard scenarios
with greater certainty about current conditions, whereas a wide range of conditions can be observed under moderate or
considerable avalanche hazard ratings. High values in the number of avalanche observations in a 72-hour period have a strong
contribution to the classification. Finally, the avalanche likelihood for persistent slab avalanches shows that the lower values
tend to have a stronger impact on the classification.

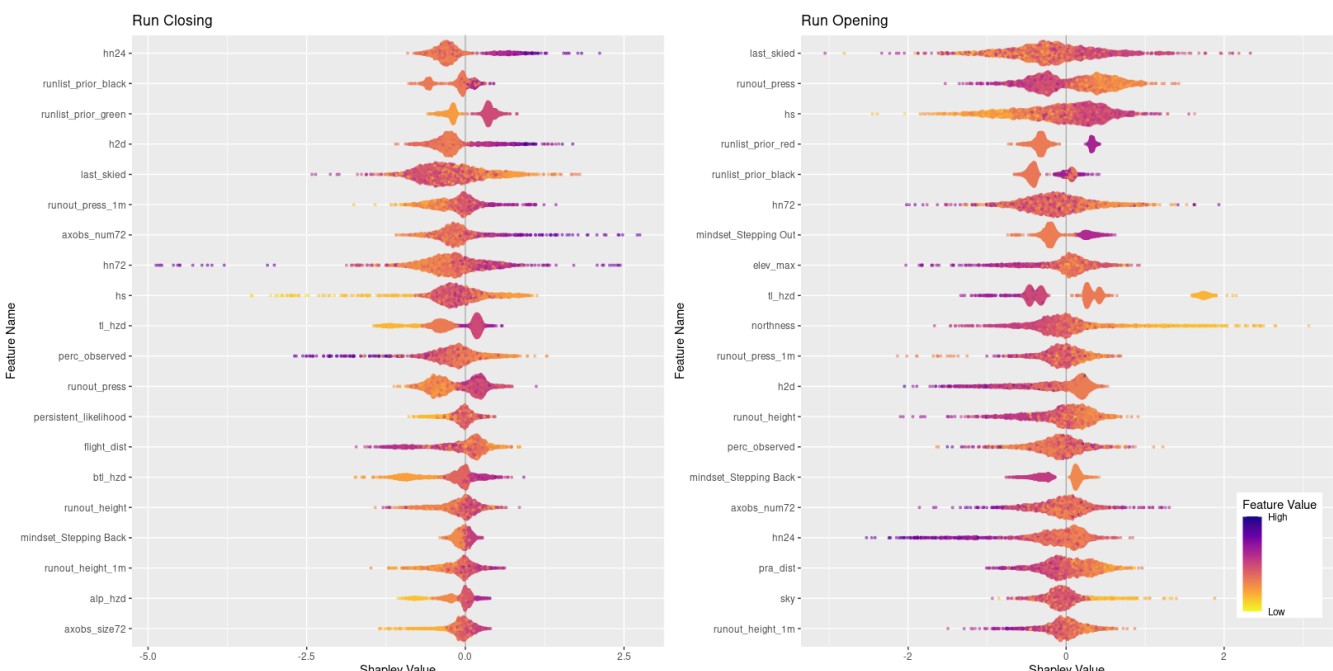



**Figure 7: SHAP plots for transition classes 'closing' and 'opening'. Features are ordered on the y-axis by their feature importance for each individual class. Negative SHAP values indicate that features are associated with that class being less likely while positive values indicate that the class is more likely. Color coded points show the relative values of the features with yellows indicating low**
**values and purple high values. Note that the order of features on the y-axis and x-axis range differs for the two plots.**

**Feature importance for transition classes**

Looking at the SHAP values for *closing* and *opening* specifically can provide additional insights about how individual features and feature values contribute to these particular decisions (Figure 7). To simplify interpretation, we removed the prior run list
feature that corresponds to the current run code (i.e., prior run list red for *closing* class), which has the highest feature importance for both classes since to open or close the run must change status from the day prior. Fifteen out of the 20 most important features are shared by both response classes, although the order of importance differs between the two classes. The 15 shared features include a mix of current conditions, terrain characteristics, and operational considerations. In general, the relationship of feature values and SHAP values is reversed for opening versus closing runs. For example, high feature values
for the height of new snow in 24 hours (*hn24*) contributes strongly to run closing (the more new snow, the more likely a run gets closed), whereas lower values of *hn24* are more important for run opening, as expected. However, the overall importance of *hn24* for opening runs is much lower than for closing runs as indicated by the difference in feature importance (rank 17 versus rank 1).

The additional five features that are only included in the top 20 features for run closing are the likelihood of persistent
avalanches, flight distance to access the run, below treeline and alpine avalanche hazard ratings, and maximum avalanche observation size over 72 hours. Low feature values of persistent avalanche likelihood align with negative SHAP values indicating that when persistent avalanches are less likely runs are less likely to be closed. Longer flight distances appear to also have a negative contribution to runs being closed, which may indicate that runs that are further from the lodge tend to change from open to closed less frequently. Avalanche hazard rating for all elevation bands have a similar distribution of
feature values and SHAP values, with lower ratings leading to less runs closing and higher ratings leading to more runs closing. Similarly, field observations of smaller avalanches contribute less to the decision to close runs than observations of larger avalanches.

The features that are only included in the top 20 for runs opening are the strategic mindset *stepping out*, maximum elevation of the run, degree of northness, total distance in PRA, and percentage of sky cover. When the guides strategic mindset is
*stepping out* runs are more likely to open, which is unsurprising but also encouraging as their stated mindset corresponds to patterns in their run list practices. Run elevation and aspect seems to contribute more to the decision to open a run for low elevation and more southerly runs than for more northerly or high elevation runs. The percentage of a run that is within PRA has the expected of effect, with lower values contributing to runs opening more heavily. Finally, runs tend to open more when





the percentage of sky cover is low. This is likely due to increased access to a larger portion of the CMH Galena tenure due to greater visibility and flying conditions from stable weather. These differences reveal some of the unique patterns identified by the XGB model in the decision-making drivers that impact the run list coding process.

**Feature importance for status quo classes**

The overall factors that have the largest contribution to runs staying open is their exposure to avalanche runout zones, the overall avalanche hazard conditions, how much recent snow loading has taken place, and whether the runs are maintained using skier traffic. Within the top 20 features by feature importance for *status green* there are four different avalanche runout features, with low values in all these features having a strong contribution to runs remaining open (Appendix C). The avalanche hazard ratings for alpine, treeline, and below treeline are also included in the top 20 features with low values having a strong contribution to runs remaining open. The same can be said for the three snow loading features, where low values contribute more strongly to runs staying open. Other terrain characteristics that contribute to runs staying open are whether they are maintained by skier traffic to mitigate persistent weak layers on the surface, lower values in the mean and 95th percentile PRA size for the frequent avalanche simulation and runs with southern aspects.

The characteristics of runs that tend to remain closed are the opposite of runs that tend to remain open, with higher runout exposure and overall higher percentage of PRA. Runs that are further away tend to remain closed more often and so do runs that start at higher elevations and have a more northerly exposure. Conditions that lead to runs remaining closed include higher avalanche hazard ratings in the alpine elevation band and higher likelihood of persistent avalanches. Snow loading over a 72-hour period contributes to the decision to keep a run closed more strongly than the 24-hour or 12-hour snowfall, which play a more important role in closing a run in the first place.

When a run is coded black it is simply not considered for the day, which is not necessarily an indication that it was deemed unsafe under the current conditions. Instead, there are a wide variety of operational factors that could play into whether a run is discussed during the morning guides meeting. Our analysis reveals several features with strong contributions to black run codes that seem to have stronger ties to operational decision-making than hazard evaluation. For example, runs that are often skied tend to have stronger contribution to being coded black, which may be an indication that guides use this code to put frequently used runs on pause during uncertain conditions instead of closing them (Appendix C). This interpretation is further supported by the observation that periods of high avalanche hazard at the treeline and below treeline elevation bands also contribute strongly to runs being coded black. Similarly, periods of higher likelihood for persistent avalanches tend to contribute to runs being coded black. Other operational considerations that contribute to runs being coded black include the flight distance, with runs further away having a strong contribution to black run codes, as well as the height of snow, with low overall snowpack heights having a strong contribution to black run codes. This pattern is likely related to more runs being coded black at the beginning of the season.





### 3.3.3 XGB Performance

The XGB model has the highest overall accuracy at 93.3%, an improvement of 1.4 percentage points over the RF. The AUROC and kappa for the XGB model are 0.98 and 0.89 respectively, which are improvements of 0.01 for AUROC and 0.02 for kappa over the RF model. The sensitivity of the transition classes for the XGB model are 72.0 % for closing and 56.8 % for opening, an improvement over the RF by 1.5 and 6.6 percentage points respectively. We used the same class weights scheme for the XGB model as the RF model, with weights determined using the inverse proportion of the class frequency. Tuning the model without class weights resulted in a slightly higher overall accuracy by 0.3 percentage points. However, the improvement in sensitivity for the transition classes 'closing' and 'opening' of 6 percentage points for both classes justify using the class weights for our application. A subset of the confusion matrix results is presented in Table 1, but interested readers are referred to Appendix C for the complete output.

## 4 Discussion

The objective of this research is to better understand the decision-making process of professional guides in terms of their daily run list coding and develop models that can meaningfully capture this decision-making process to provide decision support by producing run list predictions based on past decisions. In this section we compare the relative strengths and weaknesses of the three different models, discuss the insights that each model provides into the decision-making process, and reflect on potential applications and implications for incorporating this type of predictive model into the real-world decision-making process in mechanized skiing.

### 4.1 Summary and comparison of models

While the predictive accuracy is clearly much higher for the machine learning models than the BN, there are pros and cons to both approaches. The biggest benefit of the BN is the process of manually constructing the decision-making network by working closely with domain experts to understand the nature of the decision-making process. This collaboration required considerable time and energy to drill down into the details of the decision drivers, but the insights gained from this process not only benefitted the construction of the BN model but also the curation of the datasets and selection of features for the machine learning models. The DAG that forms the foundation of the BN is a beneficial byproduct of this process, which helps to visualize the decision-making process and captures the theoretical underpinnings (Figure 4). In addition, the combination of being based on expert input and having the predicted probabilities calculated as a simple conditional probability of input nodes makes the output of the BN much more transparent and therefore possibly more trustworthy for adoption by practitioners. Even though the interviews with domain experts identified many factors that contribute to the decision-making process, the best performing BN was limited to 23 input nodes. We found that including additional input nodes beyond what is presented


in the final model (Figure 4) decreased the predictive performance and significantly increase the computation time required to
process the predictions. The increase in processing time is due to the exponential growth of the conditional probability table
for the output node as more input nodes are added (Fenton and Neil, 2019). The reason for the predictive accuracy of the BN
decreasing when including additional variables is not obvious. Many of the additional variables that we tested in the BN were
shown to be strong predictors in the machine learning approaches (e.g. *hn24, last_skied*). Two potential causes could be that
there are strong correlations between these additional input nodes and existing nodes in the DAG or that further increasing the
number arcs directly linked to the output node is creating too large of a conditional probability table, causing relatively small
sample sizes for each potential combination of input node conditions despite our relatively large overall sample size. Due to
this trade-off between computation time and predictive performance with complexity, we manually fit and tested many
versions of the BN. To select the final version, we considered both the theoretical accuracy, as determined by our domain
expert, and prediction accuracy to arrive at a relatively simple final model. While the BN is a meaningful representation of the
high-level decision-making process, the fact that it only includes roughly half as many features compared to the machine
learning approaches may prevent the BN from capturing subtle patterns in the decision-making process and therefore contribute
to lower overall performance.

Both machine learning approaches performed better than the BN in terms of predictive performance in all accuracy metrics.
The advantage of the machine learning models was greatest in the sensitivity of the transition classes *closing* and *opening*,
with a roughly twofold increase in the percentage of cases where the actual run list was a transition class correctly identified.
This reveals that the machine learning models are much better at capturing the conditions and terrain characteristics of runs
which are likely to transition from closed to open or vice versa. The cause of the machine learning models higher skill in the
transition cases is likely due to multiple factors, including the increased number of features in the models, the inclusion of
class weights in the model fitting process which intentionally penalizes errors in the transition cases more severely, the ability
of decision tree models to naturally integrate all types of interactions between features, and the greater complexity of the
machine learning models being able to identify more subtle patterns in the data.

Between the two machine learning models the XGB showed higher performance across all accuracy metrics compared to the
RF, although the improvements were much smaller than the gap between the BN and the machine learning models (Table 1).
The largest difference between the RF and XGB was again in the transition classes, specifically the *opening* class where
accuracy improved by 6.6 percentage points over the RF. Since this class has the lowest sensitivity overall, these improvements
represent a substantial benefit to model performance. The improvement in accuracy for transition cases in the XGB model is
likely due to the boosting approach, which builds an ensemble of decision trees that use misclassified cases to sequentially
improve performance. Essentially the model identifies the cases where it is wrong and trains more decision trees to try and
improve the fit for those misclassified cases. Through this process the XGB model fitting can focus more training effort on
difficult to capture cases and potentially extract more subtle patterns in the decision-making process.



While the XGB model performed best with respect to all predictive accuracy metrics, this improvement comes with a cost from the additional feature engineering to prepare data as well as more complexity and computer resources required to tune model parameters. Finally, both machine learning models are much less transparent than the BN in terms of understanding the pathway to how the models produce their predictions. However, the same techniques for visualizing feature importance and

contributions of different features to the classification task can be applied to both.

In comparing the feature importance for the RF to the XGB model there are differences in the exact order of the features, but the highest ranked features are largely similar (Figure 5 & 6). In both models the run list prior is by far the most important, and the snow loading variables *hn24* and *hn72* are relatively highly ranked in both models. The avalanche hazard assessment variables are also highly ranked and include in the same order, with treeline hazard (*tl_hzd*) rating most important followed by

alpine (*alp_hzd*) and below treeline (*btl_hzd*). Both models rank the percent of tenure observed (*perc_observed*) and total number of avalanche observations in 72 hours (*axobs_num72*) as the most important features related to field avalanche observations. Terrain characteristics related to avalanche runout are highly ranked in both models, however the XGB model ranks the runout pressure features for the frequent and large scenarios (*runout_press, runout_press_1m*) as most important whereas the RF ranks the runout height features for frequent and large scenarios (*runout_height, runout_height_1m*) as most

important. The primary *aspect* for the RF and *northness* for the XGB both approximate the impact of solar radiation and are relatively highly ranked by feature importance. Although the RF ranks *aspect* as the most important terrain characteristic, whereas *northness* is the fourth highest ranked terrain characteristic in the XGB model. In terms of operational features, the flight distance (*flight_dist*) is highly ranked in both models.

There are several notable differences in features importance between the two machine learning models. First, the XGB model

ranks the number of days since the run was last skied (*last_skied*) as 4th overall, where it is ranked 36th in the RF model. The SHAP value plot in Figure 6 shows that high values of *last_skied* have a strong contribution to the overall classification, which may not be captured in the RF model. In contrast strategic mindset (*mindset*) is the second most important feature in the RF model and it is not included in the top 20 features of the XGB model. This is likely due to the fact the strategic mindset was dummy coded for the XGB model, so instead of determining the feature importance in aggregate across all levels of *mindset*

the XGB model considers the importance of each individual level of the *mindset* feature. The most important levels of *mindset* for the XGB model are *stepping out* and *stepping back*, which are included in the top 20 features for the class specific SHAP plots for opening and closing (Figure 7). Despite these notable differences, the fact that the feature importances are broadly similar for both machine learning models points to consistency in the ability of models to detect patterns in the guide's decision-making process.





### 4.2 Insights about decision-making process

Each of the models presented in this manuscript offer different insights into the decision-making process of professional guides. The BN illustrates the decision-making process as perceived by an expert guide and describes the essential factors that are considered when generating run lists. This has the benefit of being directly vetted by domain experts, but the predictive accuracy of such a model is likely limited by its relative simplicity compared to the real-world process. Adding more features and arcs quickly makes the model difficult to manage and understand. However, it is possible that the accessibility and transparency of this model could be most beneficial as a tool for training new guides because it illustrates the factors considered in the run list coding process and the relationships between the different factors, which could assist newer guides in developing a mental model that is in line with the past decisions of the operation. In addition, the BN model is likely more generalizable to other operations because it captures the decision process at a higher level of abstraction.

In contrast, the machine learning approaches can identify subtle patterns in the real-world data and determine which factors have the strongest relationships with past guiding decisions. The notably higher accuracy of the machine learning model predictions supports the complexity of the real-world decision-making process, with a multitude of factors impacting daily decision-making practices beyond what is possible to capture in a manually defined decision-making model. The complexity of the models required to capture their decision-making process to the best of our ability given the data available is a testament to the complex and dynamic environment that guides operate in.

Interpreting the output of the machine learning models using feature importance and SHAP value plots reveals that the patterns identified by the models seem to align with practical decision-making patterns. The relatively consistent ranking of feature importance between the RF and XGB models indicate that the models are not detecting spurious relationships, but homing in on specific factors that impact decisions under a particular set of conditions. By diving deeper into the contributions of model features for specific run list codes we can see that the factors that impact the decision-making process differ during static periods versus transition periods when runs are more likely to open and close.

### 4.3 Implications for development of decision support tools

The ultimate goal of our research is to develop meaningful and practical tools that have the potential to be integrated into the real-world decision-making process of professional guides. While the academic value of each of these models is laid out in the manuscript, there remains a question of whether guides on the ground will trust the output enough so that they can add value to the guiding operation. A key consideration along these lines is the transparency of the model and how the output compares to the guides lived experience. The BN stands above the machine learning approaches in terms of transparency and being grounded in a representation of the decision-making process of real-world guides. The decision of which model to operationalize is in the hands of the people who are accepting the real-world risk that these models aim to help mitigate.





In the case of the black box machine learning models, one approach to instil confidence in their predictions could be to front load the transparency onto the features used as input to fit the models. If guides have confidence in the data that goes into the models and know that they are trained and evaluated against their own real-world decisions, then the lack of exact details of how the predictions are made may become less important. For the models presented here, much of the data originates from guides on the ground. Field observations, hazard assessments, and operational logistics are all extracted directly from guides

or records of guiding operations which represent their perspective and past decisions. Our avalanche terrain modelling approach was originally tuned against the input of local guides (Sykes et al., 2022), and we could improve transparency and confidence in the terrain model data by providing a web mapping platform for guides to interact with the various terrain layers and gain an intuitive and personal understanding of their strengths and weaknesses.

   To apply the models in day to day guiding operations, one approach would be to use the run list classification to populate a

web map, with runs color coded according to the run list status the model predicts is most likely based on the current conditions (Figure 8). This would give the guides easy access to the model output without requiring technical knowledge to interpret the model predictions. Using the model predictions as a post assessment after their morning run list meeting could highlight cases where the guides are making decisions that are opposed to what the historic data from their operation indicates. Additional information captured in the terrain characterization, such as the GPS tracks or clustering results, PRA model output, and runout

model output, could be presented for individual runs to further help guides be on the same page about the avalanche hazard potential of the terrain in their tenure.

   A potential drawback of using predictive models as decision support tools is the potential to bias guides by revealing the predicted run list status before they have created their manual assessment. By anchoring their discussion with the model output first, a tendency to default to the model predictions could hamper guides likelihood of having a critical discussion about the

run list. The potential for this type of unintended consequence of adopting decision support tools is a real concern, and careful consideration of how to apply these tools should include both developers of the tools as well as experienced guides and operation managers.

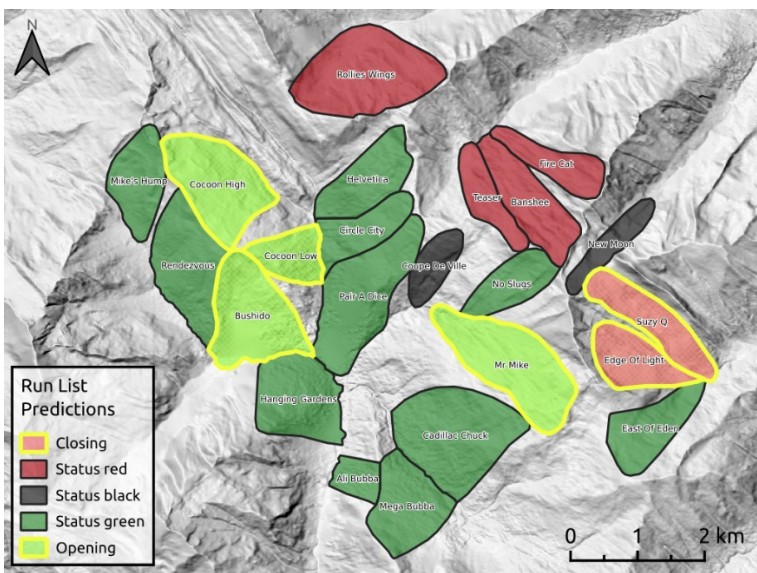

**Figure 8: Example visualization of run list codes based on predictions of decision-making tools. Runs are color coded by run list**
**prediction status.**

### 4.4 Limitations

While our analyses offer valuable insights, there are several limitations to consider when interpreting the results. Our focus on predictive performance as the benchmark for comparing model performance might undervalue the potential applications of BN for modelling the run list coding process. Since the data set of observed run list decisions we use to evaluate performance
was not validated, it can include errors, biases and inconsistencies, as well as potentially undesirable practices. While our analyses assume that our dataset at large represents meaningful decisions, it is difficult if not impossible to assess whether they were truly "correct" decisions. This is an inherent limitation of working with human judgment datasets that do not have objective validation criteria. Under these circumstances, machine learning algorithms will inherently do a better job capturing the existing patterns in a dataset. However, decision support tools developed with BN might be theoretically more valid and
produce more desirable predictions due to their grounding in a DAG. In addition, there are a huge number of potential designs to capture the run list coding decision-making process using a DAG. Our approach is based on the expertise of one experienced guide and our best assessment of meaningful relationships within the data, however alternative design strategies may be able to better incorporate additional variables in the BN and improve predictive performance.

Furthermore, both our training and validation data sets were specific to the CMH Galena tenure, and therefore the accuracy of
the models does not apply to any other guiding tenures. Finally, our decision to convert features to numeric values for the XGB model was driven by the requirements of the model and a desire for ease of interpretation. However, differences in model





performance compared to the RF, which used the categorical variables from the BN, may reflect these differences in our feature engineering choices and not purely capture the inherent advantages of the XGB model.

## 5 Conclusions

Our research aimed to combine avalanche terrain modelling, GPS tracking data, and avalanche conditions information to examine the run coding process and explore the potential for a decision support tool for mechanized ski guides. We develop three decision support tools aimed at capturing the run list process that Canadian mechanized ski guides use to determine what terrain is available for guiding each day. To characterize the important decision-making factors, we work closely with local guides at CMH Galena to understand their process. We applied data which captures current weather and avalanche hazard

conditions, operational considerations, and terrain characteristics of each run. Weather and avalanche conditions were extracted from the records of CMH Galena from the 2015/16 winter season through the 2022/23 winter season. We utilize survey data from local guides to capture the general terrain and operational characteristics of the runs included in this study.

To represent the potential avalanche terrain hazards for each run we simulate avalanche start zones and runout zones using state of the art GIS, avalanche dynamics simulation and remote sensing methods (Bühler et al., 2022; Sykes et al., 2022). The

simulations represent two different avalanche scenarios, a frequent scenario aimed at smaller magnitude avalanche events that are regularly encountered and a large scenario aimed at capturing conditions where persistent weak layers cause larger and more connected avalanches. To extract the avalanche terrain data and apply it to the decision-making models we used GPS tracks collected from guides over seven seasons to determine the portion of each run where guides regularly travel. On runs that are heavily used, we apply a clustering approach to determine the most conservative line within the run based on terrain

characteristics and pickup and landing locations to further refine the portion of the terrain data used in the decision-making models.

The three decision-making models were fit using Bayesian Network (BN), Random Forest (RF), and Extreme Gradient Boosting (XGB) approaches. The BN was built manually in close collaboration an experienced guide and is based on the theoretical real world decision-making process. The RF and XGB were fit on an expanded set of features and were each tuned

to address the class imbalance in the run list classification and to optimize the parameters of the models. Overall, the XGB model demonstrates the highest predictive performance, with an overall accuracy of 93.3 % and an area under the receiver operating curve of 0.98. All three models struggled to precisely capture cases where the run list status changed from open to closed or vice versa, with the XGB having the highest sensitivity for these classes at 72.0 % for runs closing and 56.8 % for runs opening.

While the present research represents a substantial step towards the design of practical decision support tools from operational datasets, a thorough understanding of the practical applications and consideration of unintended side effects is key to address



before operationalizing predictive models. Hence, future research should focus on how decision support tools such as the models presented in this manuscript can be applied in a meaningful way to support operational decision-making. Based on the methods developed in this manuscript, expanding the decision support tools to additional operations would be a natural next

step. However, one of the biggest hurdles to applying these methods in Canada is the relative lack of high-resolution digital elevation models (DEM). Recent development in automated Avalanche Terrain Exposure Scale (ATES) mapping (Sykes et al., 2024; Toft et al., 2024) could provide a low-cost alternative to characterize avalanche terrain hazard without the need to invest in development of high-resolution DEM data.

While the target of this research was decision support tools for mechanized guiding operations, the methods developed and

lessons learn could be adapted to a wide variety of assessment and decision-making tasks in the avalanche safety field. One key takeaway from this study is the importance of working closely with domain experts to develop decision support tools. A thorough understanding of the decision-making context and perspective of real-world practitioners is essential for meaningfully developing data sets that can capture the essential features of the decision process and for creating informed methods to evaluate predictive models. One main challenge in developing decision support tools that truly add operational value is the

requirement of large data sets which capture multiple seasons and contain a variety of avalanche conditions. We encourage operations who are interested in incorporating decision support tools into their daily practices to invest in the curation of high-quality operational records that capture the essential factors for their own decision-making processes.

**Code and data availability**

The code and data to reproduce the methods used in this paper is available at https://doi.org/10.17605/OSF.IO/6DHMX (Sykes

et al., 2024).

**Author contributions**

PH developed the infrastructure and collaboration to capture GPS tracking and operational data. JS led the preparation of GPS tracking and operational data for research application with substantial support from PH. JS led the development of avalanche terrain models with YB providing guidance and processing RAMMS simulations. RA provided operational insight for

avalanche scenarios and development of decision-making models. JS created the Bayesian network in collaboration with RA, with support from PH and PM. PH developed the GPS clustering methods, with JS and RA applying them to the study area and validating the output. JS developed the machine learning models with support from PM. JS prepared the manuscript with support from PH. All authors contributed to the final manuscript.



**Competing interests**

At least one of the (co-)authors is a member of the editorial board of Natural Hazards and Earth System Sciences. The authors have no other competing interests to declare.

**Acknowledgement**

The study area for this research is located on ancestral and unceded territories of the Secwépemc, Ktunaxa, Sinixt, and Okanagan First Nations. This research has been supported by the Government of Canada Natural Sciences and Engineering

Research Council via the NSERC Industrial Research Chair in Avalanche Risk Management at Simon Fraser University. The industry partners include Canadian Pacific Railway, HeliCat Canada, Mike Wiegele Helicopter Skiing, and the Canadian Avalanche Association. The research program receives additional support from Avalanche Canada and the Avalanche Canada Foundation.

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



## Appendix A – Variable Table

**Table A1: Description of variables included in decision support models. Variable distributions are shown with the mean and maximum values labelled for numerical variables and all classes labelled for categorical variables.**

| Full name | Abbrev. | Description | Nature and distribution | BN | RF | XGB |
|---|---|---|---|---|---|---|
| *Terrain characteristics of ski run* | | | | | | |
| Probable release area max size – frequent scenario | pra | 95th percentile of start zone polygon size of the DEM pixels covered by the GPS tracks of a run. Represents exposure of run to largest potential start zones during periods with smaller avalanches. | PRA Polygon size in m² | Categorical sml: 0-10k m² med: 10k-15k m² lrg: 15k-20k m² vlrg: > 20k m² | Same as BN | Original numerical value |
| Probable release area max size – large scenario | pra30y | 95th percentile of start zone polygon size of the DEM pixels covered by the GPS tracks of a run. Represents exposure of run to largest potential start zones during periods with larger avalanches. | PRA Polygon size in m² | Not included | Categorical sml: 0-10k m² med: 10k-15k m² lrg: 15k-20k m² vlrg: > 20k m² | Original numerical value |
| Probable release area mean size – frequent scenario | pra_mean | Mean of start zone polygon size of the DEM pixels covered by the GPS tracks of a run. Represents average size of potential start zones on a run during periods with smaller avalanches. | PRA Polygon size in m² | Not included | Categorical vsml: 0-1.5k m² sml: 1.5k-3k m² med: 3k-4.5k m² lrg: 4.5k-8k m² vlrg: > 8k m² | Original numerical value |
| Probable release area mean size – large scenario | pra30y mean | Mean of start zone polygon size of the DEM pixels covered by the GPS tracks of a run. Represents average size of potential start zones on a run during periods with larger avalanches. | PRA Polygon size in m² | Not included | Categorical vsml: 0-1.5k m² sml: 1.5k-3k m² med: 3k-4.5k m² lrg: 4.5k-8k m² vlrg: > 8k m² | Original numerical value |
| Probably release area percent of ski run | pra_perc | Proportion of DEM pixels covered by the GPS tracks on a run that are within probable release areas. | Numeric value 0 to 1 | Categorical low: 0 – 0.25 med: 0.25 – 0.4 high: 0.4 – 0.55 vhigh: > 0.55 | Same as BN | Original numerical value |
| Runout max depth – frequent scenario | runout height | 95th percentile of RAMMS runout height of the DEM pixels covered by the GPS tracks of a run. Represents exposure of run to avalanche runout deposition zones during periods with smaller avalanches. | Runout depth in m | Categorical 0 -1 m 1 - 1.5 m 1.5 - 2 m >2 m | Same as BN | Original numerical value |



| | | | | | | |
|---|---|---|---|---|---|---|
| Runout max depth – large scenario | runout height_1m | 95th percentile of RAMMS runout height of the DEM pixels covered by the GPS tracks of a run. Represents exposure of run to avalanche runout deposition zones during periods with larger avalanches. | Runout depth in m | Not included | Categorical 0 – 2.5 m 2.5 – 3.0 m 3.5 – 4.5 m > 4.5 m | Original numerical value |
| Runout max impact pressure – frequent scenario | runout press | 95th percentile of RAMMS runout impact pressure of the DEM pixels covered by the GPS tracks of a run. Represents potential avalanche runout size during periods with smaller avalanches. | Impact pressure in kPa | Categorical low: 0 – 50 kPa mod: 50 – 100 kPa high: 100 – 150 kPa vhigh: > 150 kPa | Same as BN | Original numerical value |
| Runout max impact pressure – large scenario | runout press_1m | 95th percentile of RAMMS runout impact pressure of the DEM pixels covered by the GPS tracks of a run. Represents potential avalanche runout size during periods with larger avalanches. | Impact pressure in kPa | Not included | Categorical low: 0 – 100 kPa mod: 100 – 250 kPa high: > 250 kPa | Original numerical value |
| Forested percent of ski run | forest_perc | Proportion of DEM pixels covered by the GPS tracks on a run that are within forested areas. | Numeric value 0 to 1 | Categorical 0 - 0.25% 0.25 - 0.50% 0.50 - 0.75% 0.75 - 1.0% | Same as BN | Original numerical value |
| Slope incline max | slope | 95th percentile of slope incline of the DEM pixels covered by the GPS tracks of a run. Represents the steepest terrain within the run. | Slope incline degrees | Categorical 0 - 35° 35 - 40° 40 - 45° 45 - 100° | Same as BN | Original numerical value |
| Aspect most frequent | aspect | The most frequently occurring (mode) cardinal aspect of the DEM pixels covered by the GPS tracks of a run. Represents the most prominent aspect of the run. | Cardinal direction | Not included | Categorical north: 315 - 45° east: 45 - 135° south: 135 - 225° west: 225 - 315° | Not included |
| Northness mean | northness | The average of the northness values for each DEM pixel covered by the GPS tracks of a run. Represents to degree of northern (1) versus southern (-1) exposure. | Numerical value -1 to 1 | Not included | Not included | Original numerical value |
| Elevation bands | elev | The list of elevation bands containing at least 10 % of the DEM pixels covered by the GPS tracks of a run. Represents the general elevation characteristics of the run. | Elevation bands covered | Categorical alp, tl alp, tl, btl tl, btl btl | Same as BN | Not included |
| Alpine percent of run | perc_alp | Percentage of DEM pixels covered by GPS tracks above 2250m. Represents the degree of exposure to alpine avalanche conditions for the run. | Numerical value 0 to 1 | Not included | Not included | Original numerical value |



| | | | | | | | |
|---|---|---|---|---|---|---|---|
| Treeline percent of run | perc_tl | Percentage of DEM pixels covered by GPS tracks between 1850 to 2250 m. Represents the degree of exposure to treeline avalanche conditions for the run. | Numerical value 0 to 1 | Not included | Not included | Original numerical value |
| Below treeline percent of run | perc_btl | Percentage of DEM pixels covered by GPS tracks below 1850 m. Represents the degree of exposure to below treeline avalanche conditions for the run. | Numerical value 0 to 1 | Not included | Not included | Original numerical value |
| Elevation minimum | elev_min | Lowest elevation DEM pixels covered by GPS tracks. | Elevation in m | Not included | Not included | Original numerical value |
| Elevation max | elev_max | Highest elevation DEM pixels covered by GPS tracks. | Elevation in m | Not included | Not included | Original numerical value |

***Nature of snow and avalanche conditions***

| | | | | | | |
|---|---|---|---|---|---|---|
| 72 hour new snow | hn72 | Total snowfall in cm over the past 3 days based on afternoon field observations. | Snowfall in cm | Categorical 0 cm 1 - 15 cm 15 - 30 cm 30 - 50 cm 50 - 100 cm | Same as BN | Original numerical value |
| 24 hour new snow | hn24 | Total snowfall in past 24 hours based on afternoon field observations. | Snowfall in cm | Not included | Categorical 0 cm 1 - 5 cm 5 - 15 cm 15 - 30 cm 30 - 50 cm | Original numerical value |
| 12 hour new snow | h2d | Total snowfall over past 12 hours based on morning lodge weather observations. | Snowfall in cm | Categorical 0 cm 1 - 5 cm 5 - 15 cm > 15 cm | Same as BN | Original numerical value |
| Total snowpack height | hs | Total snowpack height as measured in the field in a representative treeline location. | Snow height in cm | Not included | Categorical 0 - 150 cm 150 - 200 cm 200 - 250 cm 250 - 300 cm > 350 cm | Original numerical value |



| | | | | | | |
|---|---|---|---|---|---|---|
| Wind speed | wind | Average wind speed based on afternoon field observations. | Categorical wind speed | Not included | Categorical calm: 0 km/h light: 1 - 25 km/h mod: 26 - 40 km/h strong/extreme: > 40 km/h | Included as ordered factor |
| Precipitation rate | precip | Rate of precipitation based on afternoon field observations. | Categorical precip rate | Not included | Categorical none light: S – 1, RV, RL moderate: S1, RM heavy: ≥ S2, RH | Included as ordered factor |
| Sky cover percentage | sky | Portion of sky covered in clouds based on afternoon field observations. | Categorical sky cover | Not included | Categorical CLR: no clouds FEW: < 2/8 clouds SCT: 2/8 - 4/8 clouds BKN: 4/8 - 8/8 clouds OVC: complete clouds X: sky obscured | Included as ordered factor |
| Time of season | season | General time period of winter season based on date. The represents operational and snowpack differences across the season. | Categorical season | Categorical erly win: Nov 15 – Jan 14 mid win: Jan 15 – Feb 14 erly spring: Feb 15–Mar 14 spring: Mar 15 – Apr 15 | Same as BN | Included as ordered factor |
| Observed avalanche size 72 hours | axobs size72 | Maximum size of avalanches observed in the tenure over the past 3 days. | Avalanche destructive size | Categorical: no observations D1 - D1.5 D2 - D2.5 ≥ D3 | Same as BN | Included as ordered factor |
| Observed avalanche size 1 week | axobs sizeweek | Maximum size of avalanches observed in the tenure over the past week. | Avalanche destructive size | Categorical: no observations D1 - D1.5 D2 - D2.5 ≥ D3 | Same as BN | Included as ordered factor |
| Observed avalanche number 72 hours | axobs num72 | Total number of avalanches observed in the tenure over the past 3 days. | Total number of avalanches | Categorical 0 1 - 5 5 - 10 > 10 | Same as BN | Included as ordered factor |
| Persistent avalanche likelihood | axlkp | Likelihood rating for persistent or deep persistent avalanches based on guide hazard assessment. | Likelihood rating | Categorical none unlikely possible likely very likely | Same as BN | Included as ordered factor |

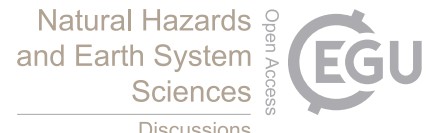

| Persistent avalanche size | axszp | Potential size of persistent and deep persistent avalanches based on guide hazard assessment. | Avalanche destructive size | Categorical none D2 – 2.5 D3 – 3.5 ≥ D4 | Same as BN | Included as ordered factor |
|---|---|---|---|---|---|---|
| Avalanche likelihood | axlk | Likelihood rating for non persistent avalanches based on guide hazard assessment. | Likelihood rating | Categorical possible likely very likely almost certain | Same as BN | Included as ordered factor |
| Avalanche size | axsz | Potential size of non persistent avalanches based on guide hazard assessment. | Avalanche destructive size | Categorical D1 – 1.5 D2 – 2.5 D3 – 3.5 ≥ D4 | Same as BN | Included as ordered factor |
| Daily max avalanche hazard | axhzd | Maximum avalanche hazard rating of the elevation bands included in the run. | Avalanche hazard rating | Categorical low moderate considerable high extreme | Not included | Not included |
| Alpine avalanche hazard | alp_hzd | Daily avalanche hazard rating for alpine elevation band. Based on daily guide hazard assessment. | Avalanche hazard rating | Not included | Categorical low moderate considerable high extreme | Included as ordered factor |
| Treeline avalanche hazard | tl_hzd | Daily avalanche hazard rating for treeline elevation band. Based on daily guide hazard assessment. | Avalanche hazard rating | Not included | Categorical low moderate considerable high extreme | Included as ordered factor |
| Below treeline avalanche hazard | btl_hzd | Daily avalanche hazard rating for below treeline elevation band. Based on daily guide hazard assessment. | Avalanche hazard rating | Not included | Categorical low moderate considerable high | Included as ordered factor |
| Strategic mindset | mindset | Daily strategic mindset determined through consensus at the daily guides meeting. | Strategic mindset | Not included | Categorical High alert Entrenchment Initial assessment Assessment Maintenance Reassessment Status quo Stepping back Stepping out Spring diurnal Open season | Included as dummy coded factor |



*Operational characteristics of ski run*

| Name | Code | Description | Plot | Col A | Col B | Col C |
|---|---|---|---|---|---|---|
| Run list code | runlist | Run list code for the run as determined at the daily guides meeting. | Run list status | Categorical: closing, status black, status red, status green, opening | Same as BN | Same as BN |
| Prior run list code | runlist_y | Run list code from the prior day as determined by the prior daily guides meeting. | Run list code | Categorical: black, green, red | Same as BN | Included as dummy coded factor |
| Run accessibility | access_gen | General accessibility of the run as determined by an online survey completed by CMH Galena guides. | Accessibility rating | Not included | Categorical: always, lineup, often, perfect | Included as ordered factor |
| Landing accessibility | access_land | General accessibility of the landing for the run as determined by an online survey completed by CMH Galena guides. | Landing access | Not included | Categorical: minimal, reasonable, well | Included as ordered factor |
| Exchange day | exchange | Binary variable indicating whether the day of the week is a typical day when guides and guests are exchanged. Impact operational logistics and decision-making. | Binary classification | Not included | Categorical: normal, exchange | Included as binary factor |
| Flight distance | flight | Distance from lodge to ski run by typical flight path. Impacts run accessibility and use frequency. | Distance in km | Categorical: near: 0 – 5 km, mid: 5 – 15 km, far: 15 – 25 km, vfar: > 25 km | Same as BN | Original numerical value |
| Days since last skied | last_skied | Number of days since the run was used for guiding. | Number of days | Not included | Categorical: 0 – 7, 7 – 14, 14 – 30, > 30 | Original numerical value |
| Operational role | op_role | Whether the run serves a specific role within the operation, such as a destination run or lunch run. This impacts how frequently runs are used due to operational logistics. | Binary classification | Categorical: no, yes | Same as BN | Included as binary factor |




| Percent of tenure observed | perc_obs | What percentage of the tenure was observed by guides in the field the prior day. | Percentage of tenure | Not included | Categorical none 1 – 5% 5 – 10% 10 – 25% 25 – 50% 50 – 100% | Original numerical value |
| Skier mitigation | ski_traf | Whether the operation uses skier traffic to destroy surficial weak layers prior to becoming buried. This created a modified snowpack and lowers likelihood of persistent avalanches. | Binary classification | Categorical do not maintain maintain | Same as BN | Included as binary factor |
| Ski quality | ski_quality | Guides perspective of the quality of the skiing experience on the run as determined by an online survey completed by CMH Galena guides. | Ski quality rating | Not included | Categorical poor - fair good very good - excellent | Included as ordered factor |






## Appendix B – Confusion Matrix Output

**Bayesian Network**

**Table B1: Bayesian Network confusion matrix.**

| | | Reference | | | | |
|---|---|---|---|---|---|---|
| | | Closing | Status black | Status red | Status green | Opening |
| **Prediction** | **Closing** | 431 | 230 | 118 | 391 | 106 |
| | **Status black** | 445 | 2692 | 124 | 163 | 250 |
| | **Status red** | 40 | 68 | 4347 | 162 | 627 |
| | **Status green** | 519 | 70 | 103 | 15240 | 68 |
| | **Opening** | 113 | 150 | 335 | 122 | 340 |

**Table B2: Bayesian Network overall Statistics**

| | |
|---|---|
| **Accuracy** | 0.8457 |
| **95% CI** | (0.8414, 0.85) |
| **No Information Rate** | 0.5899 |
| **P-Value [Acc > NIR]** | <2.2e-16 |
| **Kappa** | 0.7419 |
| **Mcnemar's Test P-Value** | <2.2e-16 |

**Table B3: Bayesian Network statistics by Class**

| | Closing | Status black | Status red | Status green | Opening |
|---|---|---|---|---|---|
| **Sensitivity** | 0.27842 | 0.83863 | 0.8647 | 0.9479 | 0.24443 |
| **Specificity** | 0.96713 | 0.95916 | 0.9596 | 0.9320 | 0.97216 |
| **Pos Pred Value** | 0.33777 | 0.73272 | 0.8289 | 0.9525 | 0.32075 |
| **Neg Pred Value** | 0.95700 | 0.97803 | 0.9691 | 0.9255 | 0.95988 |
| **Precision** | 0.33777 | 0.73272 | 0.8289 | 0.9525 | 0.32075 |
| **Recall** | 0.27842 | 0.83863 | 0.8647 | 0.9479 | 0.24443 |
| **F1** | 0.30524 | 0.78210 | 0.8465 | 0.9502 | 0.27744 |
| **Prevalence** | 0.05680 | 0.11778 | 0.1844 | 0.5899 | 0.05104 |
| **Detection Rate** | 0.01581 | 0.09877 | 0.1595 | 0.5592 | 0.01248 |
| **Detection Prev** | 0.04682 | 0.13481 | 0.1924 | 0.5871 | 0.03889 |
| **Balanced Acc** | 0.62278 | 0.89889 | 0.9122 | 0.9399 | 0.60829 |





**Random Forest**

**Table B4: Random Forest confusion matrix**

| | | Reference | | | | |
|---|---|---|---|---|---|---|
| | | **Closing** | **Status black** | **Status red** | **Status green** | **Opening** |
| **Prediction** | **Closing** | 905 | 77 | 0 | 334 | 62 |
| | **Status black** | 137 | 2090 | 0 | 0 | 119 |
| | **Status red** | 0 | 0 | 3935 | 0 | 430 |
| | **Status green** | 191 | 0 | 0 | 11663 | 0 |
| | **Opening** | 51 | 44 | 246 | 0 | 615 |

**Table B5: Random Forest overall statistics**

| | |
|---|---|
| **Accuracy** | 0.9191 |
| **95% CI** | (0.9153, 0.9227) |
| **No Information Rate** | 0.574 |
| **P-Value [Acc > NIR]** | < 2.2e-16 |
| **Kappa** | 0.8682 |
| **Mcnemar's Test P-Value** | NA |

**Table B6: Random Forest statistics by class**

| | **Closing** | **Status black** | **Status red** | **Status green** | **Opening** |
|---|---|---|---|---|---|
| **Sensitivity** | 0.70483 | 0.9453 | 0.9412 | 0.9722 | 0.50163 |
| **Specificity** | 0.97589 | 0.9863 | 0.9743 | 0.9785 | 0.98267 |
| **Pos Pred Value** | 0.65675 | 0.8909 | 0.9015 | 0.9839 | 0.64331 |
| **Neg Pred Value** | 0.98059 | 0.9935 | 0.9851 | 0.9631 | 0.96936 |
| **Precision** | 0.65675 | 0.8909 | 0.9015 | 0.9839 | 0.64331 |
| **Recall** | 0.70483 | 0.9453 | 0.9412 | 0.9722 | 0.50163 |
| **F1** | 0.67994 | 0.9173 | 0.9209 | 0.9780 | 0.56370 |
| **Prevalence** | 0.06144 | 0.1058 | 0.2001 | 0.5740 | 0.05866 |
| **Detection Rate** | 0.04330 | 0.1000 | 0.1883 | 0.5581 | 0.02943 |
| **Detection Prev** | 0.06594 | 0.1123 | 0.2089 | 0.5672 | 0.04574 |
| **Balanced Acc** | 0.84036 | 0.9658 | 0.9577 | 0.9754 | 0.74215 |





## Extreme Gradient Boosting

**Table B7: XGB confusion matrix**

| | | Reference | | | | |
|---|---|---|---|---|---|---|
| | | **Closing** | **Status black** | **Status red** | **Status green** | **Opening** |
| **Prediction** | **Closing** | 901 | 50 | 0 | 153 | 50 |
| | **Status black** | 74 | 2058 | 0 | 0 | 102 |
| | **Status red** | 0 | 0 | 3990 | 0 | 378 |
| | **Status green** | 236 | 0 | 0 | 11844 | 0 |
| | **Opening** | 40 | 39 | 287 | 0 | 696 |

**Table B8: XGB overall statistics**

| | |
|---|---|
| **Accuracy** | 0.9326 |
| **95% CI** | (0.9291, 0.9359) |
| **No Information Rate** | 0.5741 |
| **P-Value [Acc > NIR]** | < 2.2e-16 |
| **Kappa** | 0.8891 |
| **Mcnemar's Test P-Value** | NA |

**Table B9: XGB statistics by class**

| | **Closing** | **Status black** | **Status red** | **Status green** | **Opening** |
|---|---|---|---|---|---|
| **Sensitivity** | 0.72022 | 0.95855 | 0.9329 | 0.9872 | 0.56770 |
| **Specificity** | 0.98712 | 0.99061 | 0.9773 | 0.9735 | 0.98139 |
| **Pos Pred Value** | 0.78076 | 0.92122 | 0.9135 | 0.9805 | 0.65537 |
| **Neg Pred Value** | 0.98227 | 0.99523 | 0.9826 | 0.9826 | 0.97328 |
| **Precision** | 0.78076 | 0.92122 | 0.9135 | 0.9805 | 0.65537 |
| **Recall** | 0.72022 | 0.95855 | 0.9329 | 0.9872 | 0.56770 |
| **F1** | 0.74927 | 0.93951 | 0.9231 | 0.9838 | 0.60839 |
| **Prevalence** | 0.05986 | 0.10274 | 0.2047 | 0.5741 | 0.05867 |
| **Detection Rate** | 0.04311 | 0.09848 | 0.1909 | 0.5668 | 0.03330 |
| **Detection Prev** | 0.05522 | 0.10690 | 0.2090 | 0.5780 | 0.05082 |
| **Balanced Acc** | 0.85367 | 0.97458 | 0.9551 | 0.9804 | 0.77455 |





# Appendix C – SHAP value plots

## SHAP value plots for all classification levels

**995** **Figure C1: Grid of SHAP plots for the top 32 features ranked by feature importance relative to the overall classification.**



**Figure C2: SHAP value summary plot with features ranked by feature importance relative to the overall classification on the y-axis, SHAP value on the x-axis, and the relative value of the individual features shown with color coded points.**





**SHAP value plots for run closing**




**Figure C3: Grid of SHAP plots for the top 32 features ranked by feature importance relative to the class 'closing'.**







**Figure C4: SHAP summary plot for top 20 features by feature importance for classification level 'closing'.**





**SHAP value plots for status black**


**Figure C5: Grid of SHAP plots for the top 32 features ranked by feature importance relative to the class 'status black'.**
**Figure C6: SHAP summary plot for top 20 features by feature importance for classification level 'status black'.**





**SHAP value plots for status red**

[figure: grid of SHAP plots]


**Figure C7: Grid of SHAP plots for the top 32 features ranked by feature importance relative to the class 'status red'.**




**Figure C8: SHAP summary plot for top 20 features by feature importance for classification level 'status red'.**





**SHAP value plots for status green**


Figure C9: Grid of SHAP plot for the top 32 features ranked by feature importance relative to the class 'status green'.



**Figure C10: SHAP summary plot for top 20 features by feature importance for classification level 'status green'.**





**SHAP value plots for run opening**




**Figure C11: Grid of SHAP plot for the top 32 features ranked by feature importance relative to the class 'opening'.**







Figure C12: SHAP summary plot for top 20 features by feature importance for classification level 'opening'.