# Peer review of "Development of operational decision support tools for mechanized ski guiding using avalanche terrain modelling, GPS tracking, and machine learning"

_Natural Hazards and Earth System Sciences, 2024_

## Referee Comment (RC2)

**Overall Comment:**

The manuscript is well written and presents many novel methods and findings in a fascinating application. I recommend the work be published with some minor revisions. The majority of my comments are presented with the intention of providing opportunity to improve the interpretation and impact of this important paper.

The study uses recent advances in avalanche terrain modelling, an excellent application of the RAMMS dynamic model to model terrain traps and exposure to potential avalanche hazard, and three modelling approaches used to describe the run list decision-making process employed by the CMH Galena guiding team. The accuracy of the models presented is high and indicates that these models show promise in this application. Note, Bayesian Networks and Machine Learning models is not my area of expertise, and thus, I leave the critical review of the model development and technical details to others.

The manuscript may be considered long at 60 pages. The overall quality and ease of interpretation by readers may benefit if it could be condensed during the review process if possible.

**Specific Comments:**

In my understanding, avalanche likelihood and size forecasts are often produced based on the character of the avalanche expected, sometimes described as the "Avalanche Problem". Avalanche Problems usually include a broad description of where they are expected to exist within the terrain according to elevation band (i.e. Alpine, Treeline, Below Treeline) and aspect (e.g. N- NE – E, SE, …). Given that runs have varying levels of exposure to these elevation bands and aspects, it is rationale to assume that high likelihood and size ratings (i.e. elevated avalanche hazard) may exist on some runs, but not on other runs for a given day depending on the avalanche problems. Thus, the type and location of avalanche problem is likely an important factor in run list decisions. Could the authors provide an explanation of why avalanche problems were not included in the analysis? Do the authors think that a future study would benefit from including these data?

I understand that mechanized operations often use a snow safety team that gathers snowpack data, investigates conditions on runs (e.g. snow depth, crevasses, snow quality), and conducts explosive avalanche control. Operations often send snow safety teams to gather data from runs that are close to opening (e.g. coded red, black, or yellow) and these data are often critical to run list decisions. Does the variable "last_skied" represent these snow safety investigations? Or would the investigations be included in this variable along with regular skiing of the runs with guests? If snow safety investigations were not included, could the authors provide rationale why this potentially important run list decision factor was not included? Perhaps this type of snow safety investigation is not part of the regular CMH Galena practice?

In my understanding, explosive control work is often used to reduce avalanche hazard, decrease uncertainty in avalanche hazard forecasting, and to protect key features on runs prior to skiing with guests; and hence, this work often has an impact on the run list status. Does the CMH Galena operation use explosive control? If so, is there a reason that explosive control data were not

included in the analysis and modelling? Could the authors postulate the effect on the model results if a suitable dataset representing avalanche control were included?

It appears that the variable "Runlist_prior" has a very high feature importance in both the random forest and XGB models. To reduce this effect and focus the model and analysis on the factors that may lead to change decisions in the run list, did the authors consider removing runs that are rarely coded other than green (i.e. open for guiding)? That is, while I am not intimately familiar with the runs at CMH Galena, I do understand that operations often have "regular routes" or runs that involve predominantly Simple avalanche terrain (see Avalanche Terrain Exposure Scale, Statham and Campbell, 2024) and these runs are only closed for guiding in rare extreme avalanche conditions. This means that output node for these runs is likely not sensitive to the input variables for avalanche hazard conditions. Could the model and analysis provide more insight into the relevant decision factors if the output node focused on runs that often change their status?

As far as I understand, mechanized guides often use a conditional opening coding (typically coded as yellow) where a specific condition must be met prior to opening the run. If the condition is not met, the run remains closed and is not opened for skiing. Could the authors provide rationale why this standard run coding level was not included in the analysis, and postulate on what the effect on the results would have been if it was included?

Suggest increasing the size and / or resolution of the Figures to ensure they are discernible in the final publication. Currently, many of the words and symbols in the Figures are difficult to read. Specific figures where this comment is applicable are:

- Figure 1 (inset map), Figure 2 (legends, run names), Figure 4 (variable names), Figure 6 (variable names), Figure 7 (variable names), Figure 8 (run names)

An important addition to the introduction worth including is that the terrain identified in run lists already presents a significant filter on potential terrain. That is, there is much terrain that is either not skiable, too severe, or inaccessible for some reason that prevents it from even being considered on the run list. For example, a study describing the terrain indicated on the run list from well-established operations would provide value.

Lines 36-37: Suggest re-wording this sentence. It is not clear what "avalanche terrain hazard" refers to. Avalanche terrain is often described by its overall severity. This word could be an option to replace "hazard" here and elsewhere in the manuscript.

Line 64: Is the morning run-coding meeting only 15 minutes at CMH Galena?

Line 65: Suggest adding the word "may" after "… the reasons for not discussing a run…". There are other reasons why a run may remain uncoded.

Lines 126-127: Providing a list of the state-of-the-art methods or a few examples would be helpful here.

Lines 130 – 133: Please clarify that only the most conservative line on each run was used to extract terrain characteristics used to describe the avalanche terrain on each run.

Line 155: Suggest enlarging Figure 2. The legends are difficult to discern.

Line 156: Figure caption. Suggest revising the figure caption to the following or similar: "Figure 2: Comparison of PRA polygons (upper images) and runout impact pressure (lower images) for frequent (left upper and lower) and large runout (right upper and lower) simulations. The frequent PRA and impact pressure simulations represent smaller storm snow avalanches, whereas the large PRA and impact simulations represent deeper more connected persistent weak layer avalanches." Note, the images could be labelled "a, b, c, d" and referenced accordingly.

Line 190: Figure 3. Could the authors please explain or postulate why or how there are conservative clusters of lines mixed with non-conservative lines? For example, the run "Gorilla" appears to show the black (conservative lines) mixed in similar terrain with the green coloured (non-conservative lines). The figure could be improved by explaining what the differentiation between colours represents and communicating this in the legend (i.e. why does the legend only show orange and black lines when the figure shows many colours?). Lastly, the outlines for the ski run polygons could be a different colour than black because black is used for the conservative lines.

Lines 194 – 204:

- In the reviewer's understanding, the output of PRA is probabilistic. What does the extraction of PRA values along a GPS track result in? For example, is the mean PRA value the mean of all probabilistic values that the GPS track intersects which could mean that the track averages values of 0 (non-PRA raster cells) to 1 (complete PRA raster cells)?
- Slope incline is the core factor determining PRA which means that PRA and slope incline are strongly correlated. Could the model and analysis be simplified by using only PRA rather than PRA and Slope Incline as variables? Forest cover is also a strong input for PRA determination, so a similar question as to where the value is in using both forest cover and PRA in the analysis? If the model and analysis benefits from including these correlated variables, could you please explain or postulate what this value is? Further, avalanche hazard ratings will be strongly correlated with recent avalanche activity, similar to above could the authors explain the influence of including these types of strongly correlated variables in the models (note, this is more for the general knowledge of the reviewer and does not necessarily need to be included in the manuscript)?
- Similarly, avalanche runout depth, runout velocity, and runout impact pressure are strongly correlated. Could the analysis and model be simplified by using only avalanche impact pressure as an indicator of exposure to potential avalanches? Could you clarify if avalanche runout depth is used as an indicator of terrain traps or something else?

Line 197: Are the relevant GPS tracks the "conservative cluster" tracks? If so, suggest adding this slight clarification.

Lines 209 – 210: In my understanding, the destruction of weak layers by skier traffic is highly dependent on conditions. How is this variable constant?

Lines 211 – 212: In this reviewer's understanding, ski quality of a run is highly condition dependent. How is the ski quality for a run a constant variable?

Lines 402 – 403: Suggest defining the node acronyms in the legend or caption. It becomes tedious to go back and forth from the text to the figure to determine what the nodes refer to.

Line 780: Figure 8 caption. Suggest expanding the caption to explain what the yellow outlined runs refers to. For example, is it the case that the yellow outlined red runs were green the previous day and now are changing to closed?

Line 781: Limitations. In my understanding, run list coding practices vary by operation often based on the nature of the terrain, common avalanche hazard conditions, typical guests (e.g. level of ability, preferences), and experience of the guiding team. Given that the expert guide author involved in this study has an intimate understanding of the CMH Galena run list coding practices, the discussion and limitations section would benefit from some thoughts of how well these models are actually capturing the Galena run coding decisions. Are the key factors influencing run list decisions identified and do the feature importance match the expert guide author's intuition? Are there other key factors not included (e.g. explosives, snow safety investigation, avalanche problems) that would add to the decision models? Given the expert guide likely has an awareness of the run list coding practices at other operations, the discussion and limitations section would benefit from thoughts on how the results may relate to other operations.

Appendix A: In the reviewer's understanding, likelihood of avalanches is typically communicated and assessed on five level ordinal scale (i.e. Unlikely, Possible, Likely, Very Likely, Almost Certain). Could the authors please explain why the "Persistent avalanche likelihood" variable only includes the ratings None, Unlikely, Possible, Likely, and Very Likely; whereas, the "non-persistent avalanche likelihood includes the ratings Possible, Likely, Very Likely and Almost Certain?

**Technical Corrections:**

Lines 46 – 48: Suggest a slight modification:

"While these tools can be effective for general recreationists, their simplicity - particularly their focus on the public avalanche danger rating - limits their value for more complex decision-making contexts such as professional guiding or advanced amateur recreation.

Line 127: Add "," after the word "tenure".

Line 133: Add "," after the word "tools". Or use an active voice in the sentence. Note, this sentence could be combined with a slight revision of the preceding lines to clarify the terrain description process.

Line 143: Change "at al." to "et al.".

Line 108: Replace the "," after the reference with "and".

Line 196: Please check that the word "area" is appropriate after "PRA". If using the acronym in full this would equate to "… potential release area area…".

Line 566: Change "in" to "is".

Appendix A: The labels on the histogram are not readable. Please ensure the final publication uses suitable font size for readability.

Lines 218 – 221: Suggest revising these sentences to avoid the double "In addition", however this is only a stylistic writing suggestion.

Line 577: Suggest adding the numbered heading "3.3.3".

Line 607: Suggest adding the numbered heading "3.3.4".

---

## Author Comment (AC1)

**Response to reviewer comments**

Sykes, J., Haegeli, P., Atkins, R., Mair, P., and Bühler, Y.: Development of operational decision support tools for mechanized ski guiding using avalanche terrain modelling, GPS tracking, and machine learning, Nat. Hazards Earth Syst. Sci. Discuss. [preprint], https://doi.org/10.5194/nhess-2024-147, in review, 2024.

**Anonymous Reviewer # 1**

**General Comments**

The article can be printed as is, but some minor additions and elaborations could make it even more interesting. I hope my comments are constructive.

This study focuses on developing an operational decision-support tool by combining different technologies and adopting new ones. The present study will provide new insight into avalanche decision-making. It provides a possible baseline for an approach that could assist backcountry avalanche risk management and potentially serve as a training tool.

The manuscript is well-written and easy to follow. The authors made it possible for someone with limited knowledge of machine learning to understand what has been done and how the models have been developed.

**Specific Comments**

ABSTRACT

No comments

INTRODUCTION

Line 30-35
Since this tool is developed for a mechanized ski guiding environment, I would like some information on the accident/fatality rate in this setting. This is primarily to argue for the development of the tool presented and not just to show what the technology is capable of. If these figures are unavailable, the reasoning behind or need for this kind of tool should be discussed in more detail.

> **Author response: We will add a brief statement about risk due to avalanches in mechanized guiding based on research from Walcher et al., (2019) starting on line 50.**

**Walcher, M., Haegeli, P., & Fuchs, S. (2019). Risk of Death and Major Injury from Natural Winter Hazards in Helicopter and Snowcat Skiing in Canada. Wilderness & Environmental Medicine, 30(3), 251–259. https://doi.org/10.1016/j.wem.2019.04.007**

Line 47
Consider adding an extra sentence regarding why the public avalanche danger rating/trip planning tool combination is limiting in light of tools like Skitourenguru using it to create a list of runs and assigning them a color based on their risk score. Of course, that's not your problem, but it is nevertheless interesting. For example, it could contain something on scale/information density/level of uncertainty. This might help the less informed reader to a better understanding.

> **Author response: We will add a brief statement about the scale mismatch between digital avalanche terrain maps and regional avalanche forecast information on lines 47 to 49. This is outside the scope of our application but is still relevant in the broader context of decision-support tools for snow avalanche risk management.**

Line 57
Consider adding a sentence explaining the difference between a run and a line since a slope scale assessment is mentioned later in the text.

> **Author response: We will clarify the difference between ski runs and ski lines in the context of the daily guide decision-making process on lines 57 to 60.**

METHODS

Figure 1
Consider adding a table next to Figure 1 showing the number of runs in alpine and forest and the number of low-use runs in each zone that have been included in the study.

> **Author response: Appendix A shows the distributions of many avalanche terrain characteristics for the ski runs used in this study, including elevation band and forest density. We elected to include this information in an appendix due to the relatively long page count in the existing manuscript. We will add a reference to Appendix A in the caption of Figure 1 so that interested readers can easily refer to the information.**

Line 194 – 240
Consider adding a table showing the elements/factors/variables incorporated in the model for an easier overview and to avoid re-reading too much text.

> **Author response: This is a challenging aspect of this manuscript because there are many different variables and data sources included in the research. Appendix A summarizes all the variables included in the models. Including the variable**

**name, variable abbreviation, description, distributions, and how each variable is used in each decision-support model.**

3.1.1 input nodes terrain characteristics and operational factors
Out of curiosity, did you test model performance with fever input variables? What would happen if you used only runout dept or a simpler terrain characteristic indicator like automated ATES?

**Author response: Our approach to building the decision-support models started off with a simple model that included a minimum set of variables. Overall, these initial models were less accurate than the more complex versions presented in the manuscript. We based the set of variables included in the BN on the expert opinion of local guides. For the machine learning models we also included variables that demonstrated a strong relationship with the run list coding, even if our local expert did not necessarily consider them a highly important variable for their decision-making process. One example that the guide did not consider important, but that significantly improved model performance for the machine learning approaches is the slope aspect (northness) of the run.**

**We did not test automated ATES as a terrain characteristic. Prior research from Sterchi and Haegeli (2019) has hypothesized that the ATES system is too simplistic to capture the nuances of terrain relevant for professional guiding teams. However, given recent improvements in precision and accuracy of ATES maps using automated methods, further investigation of the utility of autoATES for professional guiding is necessary but beyond the scope of this manuscript. We mention the need for further research on autoATES as a decision-support tool in the conclusion section lines 830-833.**

**Sterchi, R., & Haegeli, P. (2019). A method of deriving operation-specific ski run classes for avalanche risk management decisions in mechanized skiing. Natural Hazards and Earth System Sciences, 19(1), 269–285. https://doi.org/10.5194/nhess-19-269-2019**

Model performance in general
Again, out of curiosity. Is there a difference in overall model performance, for all three models, if one distinguishes between elevation band? For example, runs that are alpine - treeline vs runs that below treeline. I could not find anything on this.

**Author response: The distribution of elevation bands is included in all three decision-support tools presented in the manuscript. For the BN and Random Forest model the list of elevation bands included in the run are captured in the variable 'elevation bands'. For example, a run could include alpine and treeline (alp,tl) or treeline and below treeline (tl, btl) bands. This is an important variable to include because it relates directly to the relevant avalanche hazard rating for the day, which are determined for each elevation band individually. So runs that only**

**contain alpine and treeline terrain would only include avalanche hazard information from those bands. We did not calculate overall model performance for subsets of elevations bands.**

DISCUSSION

Consider adding some thoughts about the mindset feature. It can be regarded as the result of an assessment of other features. This also, to some degree, applies to the last skied (and others) as it is a result of previous assessments of terrain, weather, and snow factors. How does the model perform when these "summarising factors" are excluded?

> **Author response: We will add a discussion about including the strategic mindset features in the discussion section. We intentionally omitted mindset from the BN due to the fact that it is a very high-level summary of the guides shared approach to decision-making for the day. However, in the machine learning models we aimed to determine an upper limit for predictive performance given our data and therefore elected to include the mindset variable. The results of the SHAP value plots for the XGB model illustrate that the specific mindset categories 'stepping out' and 'stepping back' are the most relevant for run list decision-making during periods of transition (lines 717 to 724). This could explain part of the reason why the machine learning models tend to perform much better in terms of predicting transitions in the run list process compared to the BN.**

Consider including some reflection on how the transparency of the BN approach could aid the identification of unknowns in the decision-making process: could the model provide some indication on what information that has to be obtained to become more certain? And could the model provide a "level of certainty score" to the user.

> **Author Response: Due to the relative simplicity of the BN compared to the real-world decision-making process it would be a stretch to apply it is a way to identify unknowns in the current real world decision-making process. There are too many potential sources of unknowns in the avalanche context for the model to meaningfully identify specific sources.**
>
> **Producing a level of certainty score fits much better with the capabilities of the decision-support tools. We will add a brief discussion of how the models could be used in an operational context to provide a confidence score along with the most likely decision outcome in lines 765-770.**

I missed a general reflection on the question: Is it at all possible to know who is right? Neither machines nor humans can predict avalanche danger with absolute certainty because we do not know the stability of the snow in space and time with sufficient accuracy.

**Author response: This is an excellent point which is currently addressed briefly in the limitations (section 4.4). However, given the importance of this concept we will also highlight it in the conclusions section.**

**Anonymous Reviewer # 2**

**Overall**

The manuscript is well written and presents many novel methods and findings in a fascinating application. I recommend the work be published with some minor revisions. The majority of my comments are presented with the intention of providing opportunity to improve the interpretation and impact of this important paper.

The study uses recent advances in avalanche terrain modelling, an excellent application of the RAMMS dynamic model to model terrain traps and exposure to potential avalanche hazard, and three modelling approaches used to describe the run list decision-making process employed by the CMH Galena guiding team. The accuracy of the models presented is high and indicates that these models show promise in this application. Note, Bayesian Networks and Machine Learning models is not my area of expertise, and thus, I leave the critical review of the model development and technical details to others.

The manuscript may be considered long at 60 pages. The overall quality and ease of interpretation by readers may benefit if it could be condensed during the review process if possible.

**Author response: Based on the comments in this review it is clear that Reviewer 2 is intimately familiar with decision-making practices in mechanized guiding operations. Their comments are very helpful for refining sections of the writing and clarifying the model development process. However, due to the length of the existing manuscript we do not plan to include all the specific details from our responses in the updated manuscript. Where specific comments or technical corrections are recommended, we will update the manuscript. Otherwise, we will review the writing with the aim of clarifying and condensing wherever possible considering our responses to the more general comments.**

**Specific Comments**

In my understanding, avalanche likelihood and size forecasts are often produced based on the character of the avalanche expected, sometimes described as the "Avalanche Problem". Avalanche Problems usually include a broad description of where they are expected to exist within the terrain according to elevation band (i.e. Alpine, Treeline, Below Treeline) and aspect (e.g. N- NE – E, SE, …). Given that runs have varying levels of exposure to these elevation bands and aspects, it is rationale to assume that high likelihood and size ratings (i.e. elevated

avalanche hazard) may exist on some runs, but not on other runs for a given day depending on the avalanche problems. Thus, the type and location of avalanche problem is likely an important factor in run list decisions. Could the authors provide an explanation of why avalanche problems were not included in the analysis? Do the authors think that a future study would benefit from including these data?

> **Author response: Avalanche problems are certainly a critical component of the avalanche hazard assessment system. There are two main reasons why we did not include more detailed information about the type, elevation, and aspect of specific avalanche problems. First, the data available was imported from the guiding operations custom database and for the majority of the years during the study period avalanche problem information was collected in a text-based field which makes automatic extraction of consistent avalanche problem data challenging. Including aspect and elevation data for the avalanche problems would have introduced large numbers of missing data and significantly limited the breadth of conditions included in our analysis. Second, our local guide recommended that persistent and deep persistent slab avalanche problems, which are included in all models, are the only specific problems where the terrain is opened and closed in a very different way. For other more surface type avalanche problems (storm slab, wind slab, loose avalanches, etc.) the specific problem type is not relevant at this scale of the decision-making process.**

> **If higher quality data were available for the aspect and elevation of specific avalanche problems we certainly would recommend exploring those in the development of future decision-support tools.**

I understand that mechanized operations often use a snow safety team that gathers snowpack data, investigates conditions on runs (e.g. snow depth, crevasses, snow quality), and conducts explosive avalanche control. Operations often send snow safety teams to gather data from runs that are close to opening (e.g. coded red, black, or yellow) and these data are often critical to run list decisions. Does the variable "last_skied" represent these snow safety investigations? Or would the investigations be included in this variable along with regular skiing of the runs with guests? If snow safety investigations were not included, could the authors provide rationale why this potentially important run list decision factor was not included? Perhaps this type of snow safety investigation is not part of the regular CMH Galena practice?

> **Author response: This type of snow safety investigation is part of the CMH Galena practice. However, we did not have reliable data as to when these snow safety investigations took place and therefore did not include this as a model variable. To the best of our knowledge the 'last_skied' variable only accounts for runs skied during guiding operations and does not account for snow safety investigations. We will double check this for the final manuscript.**

In my understanding, explosive control work is often used to reduce avalanche hazard, decrease uncertainty in avalanche hazard forecasting, and to protect key features on runs prior to skiing with guests; and hence, this work often has an impact on the run list status. Does the CMH Galena operation use explosive control? If so, is there a reason that explosive control data were not included in the analysis and modelling? Could the authors postulate the effect on the model results if a suitable dataset representing avalanche control were included?

> **Author response: Explosive control is used at CMH Galena but in a limited capacity. Our local guide recommended that the use of explosives does not play a significant role in the run list coding in Galena. However, this could be a significant factor in other operations. If quality explosive control data is available, it would likely have a significant improvement on the model because the opening and closing of those runs would likely be much less dependent on current avalanche hazard conditions.**

It appears that the variable "Runlist_prior" has a very high feature importance in both the random forest and XGB models. To reduce this effect and focus the model and analysis on the factors that may lead to change decisions in the run list, did the authors consider removing runs that are rarely coded other than green (i.e. open for guiding)? That is, while I am not intimately familiar with the runs at CMH Galena, I do understand that operations often have "regular routes" or runs that involve predominantly Simple avalanche terrain (see Avalanche Terrain Exposure Scale, Statham and Campbell, 2024) and these runs are only closed for guiding in rare extreme avalanche conditions. This means that output node for these runs is likely not sensitive to the input variables for avalanche hazard conditions. Could the model and analysis provide more insight into the relevant decision factors if the output node focused on runs that often change their status?

> **Author response: This is an excellent point. We did consider the fact that some runs rarely open or close, and elected to include all runs with at least 10 GPS tracks collected over the study period and where survey data about the guide's perception of the run was available. Our decision was to build a more generalizable model that included a variety of terrain characteristics for the results to be more transferable to other operations. However, the reviewer may be correct that removing these runs with very infrequent changes in run list status may allow the models to better capture transition periods.**

As far as I understand, mechanized guides often use a conditional opening coding (typically coded as yellow) where a specific condition must be met prior to opening the run. If the condition is not met, the run remains closed and is not opened for skiing. Could the authors provide rationale why this standard run coding level was not included in the analysis, and postulate on what the effect on the results would have been if it was included?

**Author response: Yellow run list codes were a very small portion of our overall study because they are rarely used at CMH Galena. Therefore, we elected not to include this run list status in our models due to data limitations.**

Suggest increasing the size and / or resolution of the Figures to ensure they are discernible in the final publication. Currently, many of the words and symbols in the Figures are difficult to read. Specific figures where this comment is applicable are:

- Figure 1 (inset map), Figure 2 (legends, run names), Figure 4 (variable names), Figure 6 (variable names), Figure 7 (variable names), Figure 8 (run names)

**Author response: This is an excellent point. We are happy to adjust the size and resolution of figures to improve interpretation in the final publication. The versions included in the review manuscript were slightly smaller than typical to reduce overall file size. We will work with the Copernicus publication team to ensure that all figures and tables are legible in the final version.**

An important addition to the introduction worth including is that the terrain identified in run lists already presents a significant filter on potential terrain. That is, there is much terrain that is either not skiable, too severe, or inaccessible for some reason that prevents it from even being considered on the run list. For example, a study describing the terrain indicated on the run list from well-established operations would provide value.

**Author response: We will add a brief description to lines 50-60 to explain that the run list is already a subset of the terrain within a tenure.**

Lines 36-37: Suggest re-wording this sentence. It is not clear what "avalanche terrain hazard" refers to. Avalanche terrain is often described by its overall severity. This word could be an option to replace "hazard" here and elsewhere in the manuscript.

**Author response: We will reword this sentence to improve consistency and replace the term avalanche terrain hazard throughout the manuscript.**

Line 64: Is the morning run-coding meeting only 15 minutes at CMH Galena?

**Author response: This is a rough estimate according to our local guide. We can double check this number to verify accuracy for the final manuscript.**

Line 65: Suggest adding the word "may" after "… the reasons for not discussing a run…". There are other reasons why a run may remain uncoded.

**Author response: We will add 'may' on line 65 to clarify this description.**

Lines 126-127: Providing a list of the state-of-the-art methods or a few examples would be helpful here.

**Author response: We will add a brief explanation of the 'state-of-the-art' methods used for terrain modelling on line 126-127 and include a citation to the publication where we carried out that analysis.**

Lines 130 – 133: Please clarify that only the most conservative line on each run was used to extract terrain characteristics used to describe the avalanche terrain on each run.

**Author response: We will update this sentence to clarify that only the most conservative line was used in this analysis.**

Line 155: Suggest enlarging Figure 2. The legends are difficult to discern.

Line 156: Figure caption. Suggest revising the figure caption to the following or similar: "Figure 2: Comparison of PRA polygons (upper images) and runout impact pressure (lower images) for frequent (left upper and lower) and large runout (right upper and lower) simulations. The frequent PRA and impact pressure simulations represent smaller storm snow avalanches, whereas the large PRA and impact simulations represent deeper more connected persistent weak layer avalanches."

Note, the images could be labelled "a, b, c, d" and referenced accordingly.

**Author response: These are excellent suggestions that we will add to Figure 2.**

Line 190: Figure 3. Could the authors please explain or postulate why or how there are conservative clusters of lines mixed with non-conservative lines? For example, the run "Gorilla" appears to show the black (conservative lines) mixed in similar terrain with the green coloured (non-conservative lines). The figure could be improved by explaining what the differentiation between colours represents and communicating this in the legend (i.e. why does the legend only show orange and black lines when the figure shows many colours?). Lastly, the outlines for the ski run polygons could be a different colour than black because black is used for the conservative lines.

**Author response: There are two likely reasons that the GPS tracks from the most conservative line on 'Gorilla' overlap with other GPS tracks. First is that this is a small run with relatively few distinct line options, so the difference between the lines is more subtle. Second, is that the most conservative line likely has a different pickup or drop-off location. For Gorilla, the most conservative line has drop-off locations at the upper landing and most overlapping GPS tracks start from the lower landing. Both the terrain characteristics and the drop-off and pickup location have a bearing on the GPS clustering approach.**

**These are good suggestions to improve Figure 3 which we will adopt for the final manuscript.**

Lines 194 – 204:

In the reviewer's understanding, the output of PRA is probabilistic. What does the extraction of PRA values along a GPS track result in? For example, is the mean PRA value the mean of all probabilistic values that the GPS track intersects which could mean that the track averages values of 0 (non-PRA raster cells) to 1 (complete PRA raster cells)?

**Author response: The PRA model used in this research is not probabilistic. It uses a threshold approach to create a binary PRA raster and then converts the binary raster layer into polygons using object based image analysis. The PRA variable the reviewer mentioned represents the area of PRA polygons that the GPS track intersects. We include the 95[th] percentile and mean values for both the frequent and large PRA scenarios to capture the max size of avalanche start zones encountered as well as the average exposure across the entire run. We will add a reference to Appendix A to this section where the descriptions of the various PRA variables are included. Also, the additional description we plan to add to lines 126-127 will include a more detailed overview of the PRA model methods.**

Slope incline is the core factor determining PRA which means that PRA and slope incline are strongly correlated. Could the model and analysis be simplified by using only PRA rather than PRA and Slope Incline as variables? Forest cover is also a strong input for PRA determination, so a similar question as to where the value is in using both forest cover and PRA in the analysis? If the model and analysis benefits from including these correlated variables, could you please explain or postulate what this value is? Further, avalanche hazard ratings will be strongly correlated with recent avalanche activity, similar to above could the authors explain the influence of including these types of strongly correlated variables in the models (note, this is more for the general knowledge of the reviewer and does not necessarily need to be included in the manuscript)?
Similarly, avalanche runout depth, runout velocity, and runout impact pressure are strongly correlated. Could the analysis and model be simplified by using only avalanche impact pressure as an indicator of exposure to potential avalanches? Could you clarify if avalanche runout depth is used as an indicator of terrain traps or something else?

**Author response: As mentioned above, the PRA value included is a measurement of the size of the start zone and is not as strongly correlated with slope incline or forest density as the reviewer suggests. However, the reviewer is correct that there are many correlated variables in the models including both terrain and condition variables. We included these variables because the subtle differences are important to capturing the nuanced factors that impact decision-making. For example, the avalanche runout variable for impact pressure captures the potential destructive force of avalanches while the runout depth captures areas where**

**skiers could potentially be buried deeply (terrain trap) in an avalanche event. These variables are strongly correlated, but the interpretation of the respective variables impacts the guides decision-making process in very different ways.**

**For the avalanche hazard conditions we included higher level variables, such as avalanche hazard level, because they significantly improved the overall accuracy of the models. In the BN we account for highly correlated variables, such as recent avalanches and avalanche hazard ratings, by connecting them with arcs within the decision-making network. These arcs represent the direction and hierarchy of the avalanche condition variables within the overall decision-making process. The machine learning models do not require accounting for correlated input variables, and we evaluated which variables to include based on the predictive accuracy of the models.**

Line 197: Are the relevant GPS tracks the "conservative cluster" tracks? If so, suggest adding this slight clarification.

**Author response: We will clarify this line to improve clarity.**

Lines 209 – 210: In my understanding, the destruction of weak layers by skier traffic is highly dependent on conditions. How is this variable constant?

**Author response: At CMH Galena we collected survey data from guides which captures which runs are consistently maintained using skier traffic, which runs have the potential to be maintained but are not, and which are not suitable. We simplified this variable to only represent those runs that are actively mitigated with skier traffic (1) and those that are not (0).**

Lines 211 – 212: In this reviewer's understanding, ski quality of a run is highly condition dependent. How is the ski quality for a run a constant variable?

**Author response: This variable captures the quality of the terrain on the run for skiing and not the current snow conditions. We will add a reference to Appendix A and clarify the description of 'ski quality' in Appendix A.**

Lines 402 – 403: Suggest defining the node acronyms in the legend or caption. It becomes tedious to go back and forth from the text to the figure to determine what the nodes refer to.

**Author response: We will try adding a legend to Figure 4 to improve readability.**

Line 780: Figure 8 caption. Suggest expanding the caption to explain what the yellow outlined runs refers to. For example, is it the case that the yellow outlined red runs were green the previous day and now are changing to closed?

**Author response: We will update Figure 8 caption to better capture the color coding in the map.**

Line 781: Limitations. In my understanding, run list coding practices vary by operation often based on the nature of the terrain, common avalanche hazard conditions, typical guests (e.g. level of ability, preferences), and experience of the guiding team. Given that the expert guide author involved in this study has an intimate understanding of the CMH Galena run list coding practices, the discussion and limitations section would benefit from some thoughts of how well these models are actually capturing the Galena run coding decisions. Are the key factors influencing run list decisions identified and do the feature importance match the expert guide author's intuition? Are there other key factors not included (e.g. explosives, snow safety investigation, avalanche problems) that would add to the decision models? Given the expert guide likely has an awareness of the run list coding practices at other operations, the discussion and limitations section would benefit from thoughts on how the results may relate to other operations.

**Author response: We will revisit sections 4.3 and 4.4 with the local expert guide and elicit his opinions of the model performance. However, without the expert guide using the models on a day-to-day basis under a variety of conditions it may prove difficult to make specific evaluations of the model's performance.**

**We will also include a brief discussion of how similar the CMH Galena practices are to other operations to comment on the potential transferability of these decision-support tools.**

Appendix A: In the reviewer's understanding, likelihood of avalanches is typically communicated and assessed on five level ordinal scale (i.e. Unlikely, Possible, Likely, Very Likely, Almost Certain). Could the authors please explain why the "Persistent avalanche likelihood" variable only includes the ratings None, Unlikely, Possible, Likely, and Very Likely; whereas, the "non-persistent avalanche likelihood includes the ratings Possible, Likely, Very Likely and Almost Certain?

**Author response: These differences are due to how specific avalanche problems are included in the daily avalanche hazard assessment. Non-persistent problems are not included in the avalanche problem list if the likelihood is 'unlikely', therefore this level does not exist for non-persistent avalanche problems. The opposite is true for persistent problems, they are often included in the problem list when the likelihood is 'unlikely', but within our data set there were no cases of 'almost certain' likelihood for persistent problems. Due to the nature of the model fitting we needed to drop variable levels with no observations, so that is why the list of potential likelihood levels differs for these two variables.**

**Technical Corrections**

Lines 46 – 48: Suggest a slight modification:

"While these tools can be effective for general recreationists, their simplicity - particularly their focus on the public avalanche danger rating - limits their value for more complex decision-making contexts such as professional guiding or advanced amateur recreation.

Line 127: Add "," after the word "tenure".

Line 133: Add "," after the word "tools". Or use an active voice in the sentence. Note, this sentence could be combined with a slight revision of the preceding lines to clarify the terrain description process.

Line 143: Change "at al." to "et al.".

Line 108: Replace the "," after the reference with "and".

Line 196: Please check that the word "area" is appropriate after "PRA". If using the acronym in full this would equate to "… potential release area area…".

Line 566: Change "in" to "is".

Appendix A: The labels on the histogram are not readable. Please ensure the final publication uses suitable font size for readability.

Lines 218 – 221: Suggest revising these sentences to avoid the double "In addition", however this is only a stylistic writing suggestion.

Line 577: Suggest adding the numbered heading "3.3.3".

Line 607: Suggest adding the numbered heading "3.3.4".